# THE DISTRIBUTIONAL REWARD CRITIC ARCHITECTURE FOR PERTURBED-REWARD REINFORCEMENT LEARNING

## ABSTRACT

We study reinforcement learning in the presence of an unknown reward perturbation. Existing methodologies for this problem make strong assumptions including reward smoothness, known perturbations, and/or perturbations that do not modify the optimal policy. We study the case of unknown arbitrary perturbations that discretize and shuffle reward space, but have the property that the true reward belongs to the most frequently observed class after perturbation. This class of perturbations generalizes existing classes (and, in the limit, all continuous bounded perturbations) and defeats existing methods. We introduce an adaptive distributional reward critic and show theoretically that it can recover the true rewards under technical conditions. Under the targeted perturbation in discrete and continuous control tasks, we win/tie the highest return in 40/57 settings (compared to 16/57 for the best baseline). Even under the untargeted perturbation, we still win an edge over the baseline designed especially for that setting.

## 1 INTRODUCTION

The use of reward as an objective is a central feature of reinforcement learning (RL) that has been hypothesized to constitute a path to general intelligence (Silver et al., 2021). The reward is also the cause of a substantial amount of human effort associated with RL, from engineering to reduce difficulties caused by sparse, delayed, or misspecified rewards (Ng et al., 1999; Hadfield-Menell et al., 2017; Qian et al., 2023) to gathering large volumes of human-labeled rewards used for tuning large language models (LLMs) (Ouyang et al., 2022; Bai et al., 2022). Thus, the ability of RL algorithms to tolerate noisy, perturbed, or corrupted rewards is of general interest (Romoff et al., 2018; Wang et al., 2020; Everitt et al., 2017; Moreno et al., 2006; Corazza et al., 2022; Zheng et al., 2014).

Different reward correction methods have been proposed to enhance the performance of RL algorithms. Flow chart Fig 8 in Appendix A is to help understand the pipeline. However, many of these existing approaches rely on strong assumptions regarding perturbed rewards. For instance, the reward estimation (RE) method (Romoff et al., 2018) assumes that continuous perturbations do not impact the optimal policy, a condition satisfied in limited cases, such as when the reward undergoes an affine transformation. On the other hand, the surrogate reward (SR) method introduced in (Wang et al., 2020) can handle perturbations beyond affine transformations, but it presupposes discrete rewards and a readily inferable perturbation structure. Nevertheless, it remains unclear how to infer the confusion matrix in the general scenario where rewards, states, and actions are continuous. We provide a detailed discussion about these works and other related works in Sec. 2.

**Contributions** In this paper, we aim to address these limitations by proposing methods that can handle a wider range of perturbations. Our contributions can be summarized as follows.

- **A distribution reward critic (DRC) architecture** Inspired by recent successes in distributional reinforcement learning (Dabney et al., 2018b; Bellemare et al., 2017; Dabney et al., 2018a; Rowland et al., 2018), we propose a distributional reward critic (DRC) that models the distribution of perturbed rewards for each state-action pair using a neural network. Compared with the SR (Wang et al., 2020) which relies only on the observed reward, our approach can leverage the correlations between different state-action pairs to guide the estimation of the true rewards from perturbed ones.

Our approach also formulates the estimation as classification instead of regression, by leveraging the common observation that training on the cross-entropy loss results in better performance (Stewart et al., 2023).

- **Theoretical analysis under generalized confusion matrix.** We conduct a theoretical study of the proposed DRC under a generalized confusion matrix (GCM) perturbation model. Our GCM generalizes the confusion matrix perturbation of (Wang et al., 2020) by accommodating continuous rewards Under GCM, we show that the DRC can recover the distribution of the perturbed reward under technical conditions. Thus, DRC can recover the true rewards if mode-preserving is satisfied, i.e., the most frequently observed reward for each state-action pair is the true reward, which is also the assumption used in previous works. We provide a comparison with other approaches in Tab. 1.
- **Experimental performance.** We compare the distributional reward critic to methods from the literature for handling perturbed rewards under a wide selection of reward perturbations. Under our targeted perturbation, we win/tie (95% of the winning performance) the highest return in 40/57 sets (compared to 16/57 for the best baseline). Even under the untargeted one, we still win an edge over the baseline that is designed for that setting.

Table 1: The comparison of the methods from different perspectives. The bolded methods are introduced in this paper. Blue is a strength, and orange is a weakness. Required Information is the less the better. Under the Reward Estimator Property group, variance concerns the estimated rewards for different samples of any state-action pair $(s, a)$. Under the Required Perturbation Property group, Optimal-Policy-Unchanged means the optimal policy in expectation of the perturbed rewards stays unchanged, the details of Assumption will be discussed soon.

| Method | Required Information | | | Reward Estimator Property | | Required Perturbation Property | |
| --- | --- | --- | --- | --- | --- | --- | --- |
| | $\tilde{R}$ | $n_r$ | Reward Range | Reduces Variance | Leverages $(s, a)$ | Optimal-Policy-Unchanged | Assumption |
| Standard | ✗ | ✗ | ✗ | ✗ | ✗ | ✓ | Clean Environment |
| SR_W | ✓ | ✓ | ✓ | ✗ | ✗ | ✗ | Mode-Preserving |
| SR | ✗ | ✓ | ✓ | ✗ | ✗ | ✗ | Mode-Preserving Smooth Reward |
| RE | ✗ | ✗ | ✗ | ✓ | ✓ | ✓ | $\mathbb{E}(\tilde{r}) = \omega_0 \cdot r + \omega_1 (\omega_0 > 0, (\omega_0 \cdot r + \omega_1) \cdot r > 0)$ |
| **DRC** | ✗ | ✓ | ✗ | ✓ | ✓ | ✗ | Mode-Preserving |
| **GDRC** | ✗ | ✗ | ✗ | ✓ | ✓ | ✗ | Mode-Preserving |

## 2 RELATED WORK

Perturbation of RL systems have been extensively studied (Rakhsha et al., 2020; Huang et al., 2017; Kos & Song, 2017; Lin et al., 2017b;a; Behzadan & Munir, 2017; Pattanaik et al., 2017; Pinto et al., 2017; Choromanski et al., 2020). Closest to ours are (Romoff et al., 2018) and (Wang et al., 2020).

**Continuous Perturbation And The Reward Estimation (RE) Method** (Romoff et al., 2018) focus on variance reduction in the case of continuous perturbations that do not affect the optimal policy, i.e., maximizing the return in the perturbed reward MDP also maximizes rewards in the original MDP. This occurs, for instance, when the reward perturbation applies an affine transformation to the expected reward with a non-negative scale factor, i.e., $\mathbb{E}[\tilde{R}(s, a)] = w_0 R(s, a) + w_1$ and $w_0 > 0$. In this setting, the perturbation can slow training by increasing the reward variance. They propose a reward estimation (RE) method by introducing a reward critic that predicts $\tilde{R}(s, a)$ from $s$ and $a$ that aims to reduce this variance. This critic replaces observed rewards from the environment and is trained to predict the observed reward to minimize mean-squared error, converging to the expectation of the perturbed rewards.

**Confusion Matrix Perturbation And The Surrogate Reward (SR) Method**  (Wang et al., 2020) study the setting where the clean rewards are discretized and then perturbed by a confusion matrix $C$, where $C(i, j)$ represents the probability that the samples with rewards $r_i$ are with rewards $r_j$ after the perturbation. This perturbation leads to the changed optimal policy in expectation of the perturbed rewards because the perturbation distributions are different for different clean rewards compared with the continuous perturbation in (Romoff et al., 2018), which influences the optimal policy measured by the observed and noisy rewards. They propose a surrogate reward (SR) method by inverting the estimated/known confusion matrix and computing the predicted rewards as $\hat{R} = C^{-1} \cdot R$, where $R$ represents the vectorized, clean, and discrete rewards and $\hat{R}$ represents the vectorized and predicted rewards, replacing each perturbed reward with an unbiased estimate of the true reward because $C \cdot \hat{R} = R$. Their approach comes with a strong informational cost in the form of a known or easily estimable confusion matrix. SR can learn the matrix within a simple environment space (e.g. Pendulum), but SR_W needs to know the matrix before being applied.

**Distributional RL**  The distributional reward critic approach we propose is inspired by distributional RL (Dabney et al., 2018b; Bellemare et al., 2017; Dabney et al., 2018a; Rowland et al., 2018), where the value function is modeled distributionally rather than as a point estimate. We model the reward function as a distribution but encounter issues that do not arise in distributional RL, where ground truth rewards are directly observable. However, standard quantile regression techniques divide probability density into fixed-sized bins, rendering them unsuitable for peak identification. Instead, we use a fixed-width discretization, through which we adaptively control the granularity while extracting the mode.

**Noisy Label Learning**  We are additionally inspired by work in noisy label learning, e.g., (Song et al., 2022; Ghosh et al., 2017; Zhang & Sabuncu, 2018; Ma et al., 2020; Liu et al., 2022), which show that strong classification models can be constructed even under a large degree of label perturbation.

## 3 PROBLEM STATEMENT

Let $\langle S, A, R, P, \gamma, \beta \rangle$ be a Markov Decision Process (MDP) (Puterman, 2014), where $S$ is the state space, $A$ is the action space, $R : S \times A \to \mathbb{R}$ is the reward function, $P : S \times A \to \Delta S$ is the transition function, $\beta \in \Delta S$ is the initial state distribution, and $\gamma \in [0, 1]$ is the discount factor. We denote the state at timestep $t$ as $s_t$, the action as $a_t$, and the reward as $r_t = R(s_t, a_t)$.

### 3.1 PERTURBED REWARD MDP

We define a *perturbed reward MDP* of the form $\left\langle S, A, R, \tilde{R}, P, \gamma, d_0 \right\rangle$. This is a standard MDP except the agent perceives perturbed rewards from $\tilde{R}$ instead of true rewards from $R$. Following prior work, we assume that the distribution of the perturbed reward depends only on the true reward, i.e., $\tilde{r}_t \sim \tilde{R}(R(s_t, a_t))$. The objective is the same as in a standard MDP: we seek a policy $\pi : S \to \Delta A$ that maximizes the return, i.e., $\pi^* \in \arg\max_\pi \mathbb{E}_\pi [\sum_{t=0}^\infty \gamma^t R(s_t, a_t)]$.

### 3.2 GENERALIZED CONFUSION MATRIX PERTURBATIONS

We discretize the reward range into $n_r$ intervals, each with an equal length $\ell_r = (r_{\max} - r_{\min})/n_r$, where $r_{\min}$ and $r_{\max}$ represent the minimum and maximum rewards attainable from the environment. Each reward $r \in [r_{\min}, r_{\max})$ has a *label* $y \in Y = [0, n_r - 1]$ with $y = \lfloor (r - r_{min})/l_r \rfloor$. The confusion matrix $C$ is an $n_r$ by $n_r$ with $C(i, j) \in [0, 1]$ and $\sum_{j \in Y} C(i, j) = 1$. The *perturbed label* of sample $y$ is $\tilde{y}$, i.e., $j$ with probability $C(y, j)$. Then, we shift $r$ by the signed distance between $y$ and $\tilde{y}$, i.e., $\tilde{r} = r + D(\tilde{y}, y)$, where $D$ is the signed Euclidean distance between the interval centers $D(\tilde{y}, y) = (\tilde{y} - y) \cdot \ell_r$. We provide an example of the application of a GCM perturbation in Fig. 9 in Appendix A.

The GCM does not sparsify the input rewards, i.e., if the input rewards are continuous, the perturbed rewards will be as well. It generalizes the confusion matrix perturbation of (Wang et al., 2020), which produces a finite number of possible reward values if naively applied to a continuous reward signal.

The GCM perturbation can represent perturbations with an arbitrary PDF that is different for each interval—each row of the matrix is a perturbation PDF for rewards in that interval. Specifically, it can represent perturbations that are optimal-policy-changed, i.e., the optimal policy in expectation of perturbed rewards changes after perturbations. We term this "optimal-policy-changed perturbation", which cannot be addressed by RE (Romoff et al., 2018).

We remark that the GCM can approximate any continuous perturbation with a bounded error that diminishes with the increase of the number of intervals $n_r$ as shown in Proposition 1, whose proof is in Appendix B.1.

**Proposition 1.** *Consider a continuous perturbation model that for each reward $r \in [r_{\min}, r_{\max})$, it can be perturbed to $\bar{r} \in [r_{\min}, r_{\max})$. Our confusion matrix represents $\bar{r}$ with $\tilde{r}$ that satisfies $|\tilde{r} - \bar{r}| \leq \frac{r_{\max} - r_{\min}}{n_r}$.*

Under GCM perturbations, we have the same mode-preserving assumption as SR. With this assumption, we seek methods that can recover the perturbations breaking the optimal policy RE cannot handle. For the perturbations people care about in RE, mode-preserving still holds, leaving the possibility for our methods to work under another untargeted perturbation setting. It is because of the weak mode-preserving assumption, our methods can work well under many kinds of perturbations.

## 4 THE DISTRIBUTIONAL REWARD CRITIC

In this section, we introduce the distributional reward critic. We begin with the simpler case where the reward range and the number of intervals in the GCM are known in Sec. 4.1. Then, we study how to infer the number of intervals and the reward range in Sec. 4.2.

### 4.1 DISTRIBUTIONAL REWARD CRITIC (DRC) WITH KNOWN DISCRETIZATION

---

**Algorithm 1** Distributional Reward Critic (Known Discretization)

---

1: **procedure** UPDATE AND CRITIQUE SAMPLES
2:    **Notation:** $\tilde{y}_t$: noisy reward label; $r_{\min}$: the minimum reward; $r_{\max}$: the maximum reward; $\ell_r$: the length of each reward interval $\ell_r = (r_{\max} - r_{\min})/n_r$; reward critic $\hat{R}_\theta$
3:    **Input:** samples $\{(s_t, a_t, \tilde{r}_t)\}$
4:    **Output:** samples with predicted rewards $\{(s_t, a_t, \hat{r}_t)\}$
5:    Augment each sample with a discrete label $\tilde{y}_t = \text{floor}\big( (\tilde{r}_t - r_{\min}) / \ell_r \big)$
6:    Train $\hat{R}_\theta$ using Adam optimizer to minimize cross entropy[1] $\sum_t H(\tilde{y}_t, \hat{R}_\theta(s_t, a_t))$.
7:    For each sample, predict discrete reward label $\hat{y}_t = \arg\max_{y \in Y} \hat{R}_\theta(s_t, a_t)$ and compute predicted reward value $\hat{r}_t = \tilde{r}_t + \ell_r(\hat{y}_t - \tilde{y}_t)$
8:    Return critiqued samples $\{(s_t, a_t, \hat{r}_t)\}$

---

We propose a distributional reward critic (DRC) (Alg. 1) that views the reward critic's task as a multi-class classification problem: given input $(s, a, \tilde{r})$ predict a discrete distribution over the reward range that minimizes the cross-entropy loss, i.e., $\hat{R}_\theta : S \times A \to \Delta Y$ where $Y = (0, 1, \ldots, n_r - 1)$. Then, we select the most probable reward $\hat{r} \equiv \tilde{r} + \ell(\hat{y} - \tilde{y})$, where $\hat{y}$ is the highest probability label from $\hat{R}_\theta(s, a)$. This DRC can be inserted into any RL algorithm. When new $(s, a, \tilde{r})$ tuples are collected from the environment, they are used to train $\hat{R}_\theta$. When the algorithm requires $r$, the reward critic output $\hat{r}$ is provided instead.

**Comparison with SR and RE**  SR (Wang et al., 2020) relies solely on the observed reward, computed as $\hat{R} = C^{-1} \cdot R$, when the confusion matrix $C$ is either known or estimated. Consequently, since different samples of the same state-action pair may yield distinct perturbed rewards, the SR method might produce varying estimated rewards with significant variance. In contrast, our proposed DRC, akin to RE (Romoff et al., 2018), addresses this issue by learning a reward prediction mapping

---

[1]With slight abuse of notation, $H(\tilde{y}_t, \hat{R}_\theta(s_t, a_t))$ denotes the cross entropy between $\hat{R}_\theta(s_t, a_t)$ and the distribution with all zero probabilities except for the $\tilde{y}_t$-th being 1 (i.e., the one-hot vector version of $\tilde{y}_t$).

for each state-action pair. This approach not only ensures a consensus in reward prediction (resulting in zero variance) for different samples of a state-action pair, but also harnesses reward correlations across different state-action pairs, facilitating collaborative prediction learning. However, the RE method learns reward prediction via regression, aiming to capture the expectation of the perturbed reward. This implies that the perturbations should be affine transformations (or optimal-policy-unchanged) as discussed in Sec. 2 and Sec. 3.2. Conversely, our classification-based approach can accommodate a wider range of perturbations.

**Exact recovery of DRC under GCM** Intuitively, if the confusion matrix perturbation can be identified, the maximum probability label can be identified and thus the true reward can be recovered with no reconstruction error, regardless of the underlying reward distribution, which is the result of Thm. 1, whose proof is in Appendix B.2.

**Theorem 1.** *With a sufficiently expressive neural network and GCM perturbations, the prediction from DRC for a sample $(s_t, a_t)$ is $C(y_t, :)$ (the $y_t$-th row of $C$), which is the distribution of the perturbed reward label, as the number of samples from each state-action pair approaches infinity. Consequently, the correct reward can be exactly predicted if the perturbation is mode-preserving with known discretization.*

Note that this result applies even when the true reward distribution is continuous because the confusion matrix always shifts rewards by an integer multiple of $\ell_r$—if this multiple can be predicted, there will be no reconstruction error.

The DRC's requirement that the structure of the GCM will be met in tasks where the number of different possible reward values is finite. Following prior work, we'd like to relax these assumptions and allow the GCM structure to be learned from data and to be applicable to non-GCM rewards.

## 4.2 GENERAL DRC (GDRC) WITH UNKNOWN DISCRETIZATION

We begin with the case of an unknown number of reward intervals $n_r$. In Section 4.2.2, we explain how to handle the unknown reward range.

### 4.2.1 KNOWN REWARD RANGE, UNKNOWN NUMBER OF INTERVALS

Our primary strategy is to guess the number of intervals from DRC *differential* cross-entropy loss as the number of outputs $n_o$ varies. We begin by studying the impact of $n_o$ on the reconstruction error. Then we study how to use the cross-entropy loss to select $n_o$ to minimize reconstruction error.

**Impact of number of intervals $n_o$ on reconstruction error** When $n_o \neq a \cdot n_r (a \in \mathbb{Z}^+)$, it is no longer the case that $\text{ERROR}_r$ approaches 0 in the infinite sample limit. There is now an irreducible source of reconstruction error that is caused by misalignment of the intervals in the reward critic compared to the perturbation. The dynamics of this *misalignment error* are non-trivial—it is generally smaller when $n_o$ is larger.

To intuitively illustrate the misalignment error, we divide the range $[r_{\min}, r_{\max}]$ into $n_o$ contiguous intervals of equal length $\ell_o = (r_{\max} - r_{\min})/n_o$. In this case, the labeling set becomes $Y_o = (0, 1, \ldots, n_o - 1)$, and given a perturbed reward $\tilde{r} = r + \ell_r(\tilde{y} - y)$, we select the most probable reward $\hat{r} = \tilde{r} + \ell_o(\hat{y}_o - \tilde{y}_o)$, where $\hat{y}_o = \arg\max_{y_o \in Y_o} \hat{R}_\theta(s, a)$. Even if the network $\hat{R}_\theta$ is sufficiently expressive and is trained to give a correct prediction of $\hat{y}_o$, $\hat{r}$ is not a correct prediction of $r$ due to the misalignment between $\ell_r(\tilde{y} - y)$ and $\ell_o(\hat{y}_o - \tilde{y}_o)$. In general, when $n_o < n_r$, the misalignment becomes more pronounced as the difference between $n_o$ and $n_r$ increases. When $n_o > n_r$, the misalignment still exists, except for the case that $n_o$ is a multiple of $n_r$ wherein $\hat{y}_o - \tilde{y}_o$ is also a multiple of $y - \tilde{y}$. In Fig.1, we summarize this discussion about the impact of the

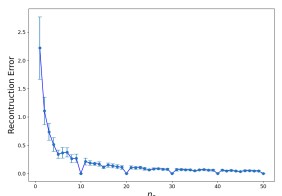

Figure 1: The reconstruction error initially decreases as $n_o$ increases, reaches 0 at $n_o = n_r$, and then oscillates.

number of intervals on reconstruction error computed as $\text{ERROR}_r(\hat{R}_\theta) = \frac{1}{|T|} \sum_{t \in T} |\hat{r}_t - r_t|$, where $T$ represents the number of samples we use DRC for prediction. We provide a detailed discussion in the Appendix C.

Under the infinite samples assumption, a large $n_o$ that is near a multiple of $n_r$ is preferred as errors in estimating $n_r$ are less costly. Without this assumption, there is a tradeoff—a large $n_o$ leads to worse overfitting because of limited samples, but a small $n_o$ leads to worse reconstruction error. We study this interplay empirically in Section 5.2. For now, we focus our attention on setting $n_o = n_r$, as it achieves zero reconstruction error, and we show cross-entropy is an accessible metric to help estimate it in the next part.

**Leveraging the training cross-entropy loss to select** $n_o$ During reward critic training, we can view the cross-entropy loss of the reward critic. We turn our attention to the dynamics of this loss as $n_o$ changes, and we will show that it can be used to estimate $n_r$.

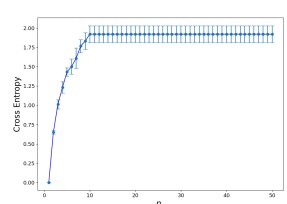

Consider a single state action pair $(s, a)$ with a reward label of $y$. Let $p_{r \to o}$ represent the true distribution of perturbed reward labels $C(y)$ discretized into $n_o$ equal length intervals, and let $p_o$ represent the reward critic distribution $\hat{R}_\theta(s, a)$ of reward labels. Recall that $H(p_{r \to o}, p_o)$ denotes the cross entropy loss between the two distribution $p_{r \to o}$ and $p_o$.

Figure 2: The minimum cross entropy of the reward critic increases as $n_o$ increases until $n_o$ reaches $n_r$.

**Theorem 2.** *Consider the infinite sample limit. When $n_o \leq n_r$, $\min_{p_o} H(p_{r \to o}, p_o)$ is non-decreasing in $n_o$. Moreover, if $P(C(y) = y') > 0$ for $\forall y, y' \in Y$, $\min_{p_o} H(p_{r \to o}, p_o)$ is an increasing function of $n_o$. When $n_o \geq n_r$, $\min_{p_o} H(p_{r \to o}, p_o) = H(p_{r \to r})$, the entropy of $p_r$.*

Refer to Appendix B.3 for the proof of Thm. 2. Fig. 2 shows the simulation results of $\min_{p_o} H(p_{r \to o}, p_o)$ as $n_o$ increases. Intuitively, the minimum cross entropy is influenced by the interval length, with larger intervals fostering simpler distributions and consequently, reduced cross entropy. This suggests that identifying when the reward critic cross entropy stops increasing as $n_o$ increases can guide identifying $n_r$.

We propose training an ensemble of reward critics $\left\{ \hat{R}_\theta^{(n_o)} \right\}$ with uniform perturbation discretizations $n_o \in N_o$. We use these critics to vote on where the rate of increase of cross entropy starts increasing. We use the critic who has received the most votes with a discount factor as the sole reward critic at each epoch. Specifically, let $\delta H^{(n_o)} = H(p_{r \to o}, p_o) - H(p_{r \to o'}, p_{o'})$, where $n_{o'}$ is the largest element in $N_o$ that is less than $n_o$. We then define the winning critic on epoch $t \leq T_{vote}$ as $\arg\min_{n_o} \{ \delta H^{(n_o)} > \delta H^{(n_{o'})} \}$. See Appendix D for the pseudocode of the GDRC method.

### 4.2.2 UNKNOWN RANGE, UNKNOWN NUMBER OF INTERVALS

For an unknown reward range and an unknown number of intervals case, we use the "voting critics" from the previous section plus an addition that updates the intervals based on the observed rewards seen so far using a streaming technique to compute percentiles (Masson et al., 2019). We create variables $r_{\text{emin}}$ and $r_{\text{emax}}$ to store the 5% percentile and 95% of the observed rewards across all samples, which excludes the influence of outlier perturbation because of long-tail noise.

## 5 EXPERIMENTAL RESULTS

In this section, we experimentally demonstrate that DRC and GDRC methods outperform existing approaches by attaining higher true rewards and exhibiting applicability across a broader spectrum of environments and perturbations. Section 5.1 introduces the algorithms, environments, and perturbations. The influence of $n_o$ on reconstruction errors and cross-entropy, in line with the discussion in Section 4.2.1, is substantiated in Section 5.2. In Section 5.3 and Section 5.4, we compare our methods with baseline methods under the confusion matrix and continuous perturbations respectively.

### 5.1 EXPERIMENTAL SETUP

**Algorithms** The methods introduced for perturbed rewards in this paper and in the baselines we consider can be applied to any RL algorithm. Thus, we compare all methods as applied to some popular algorithms such as Proximal Policy Optimization (PPO) (Schulman et al., 2017), Deep

Deterministic Policy Gradient (DDPG) (Lillicrap et al., 2015), and Deep Q Network (DQN) (Mnih et al., 2013), covering on-policy and off-policy algorithms. The methods introduced by this paper are **DRC** (with $n_r$ and reward range) and **GDRC** (without any information). For baselines, we compare to the original algorithms mentioned above, RE (Romoff et al., 2018), and SR (SR and/or SR_W) (Wang et al., 2020). Each method has its reward averaged over 10 seeds and 20 random trials after the training and +/- shows the standard error in Appendix F.

**Environments**   We first consider two simple control tasks, Pendulum and CartPole (the environments tested by SR). Then we consider some more complex Mujoco environments: Hopper-v3, HalfCheetah-v3, Walker2d-v3, and Reacher-v2 (Todorov et al., 2012) (the environments tested by RE).

**Perturbations**   We test two kinds of perturbations: GCM and the continuous perturbations. For GCM perturbation, we vary two parameters: the number of intervals $n_r$ and the noise ratio $\omega$. An $\omega$ proportion of samples in the interval containing the true reward are perturbed into any interval at random. For continuous perturbation, we test the same distributions as (Romoff et al., 2018). Gaussian noise is an additive zero-mean Gaussian distribution: $\tilde{r}_t = r_t + \mathcal{N}(0, \omega^2)$. For uniform noise, with a probability of $\omega$, the reward is sampled uniformly from $[-1, 1]$ and is unaltered otherwise. We also consider a "reward range uniform" noise, adjusting the range of the uniform distribution to $U(r_{\min}, r_{\max})$, where $r_{\min}$ and $r_{\max}$ signify the minimum and maximum reward achievable by an agent per environment.

## 5.2   IMPACT OF $n_o$ ON RECONSTRUCTION ERRORS AND CROSS-ENTROPY

Here we study the impact of $n_o$ on the reconstruction error, episode reward, and cross-entropy. Recall a theoretical zero reconstruction error is achieved whenever $n_o$ is a multiple of $n_r$.

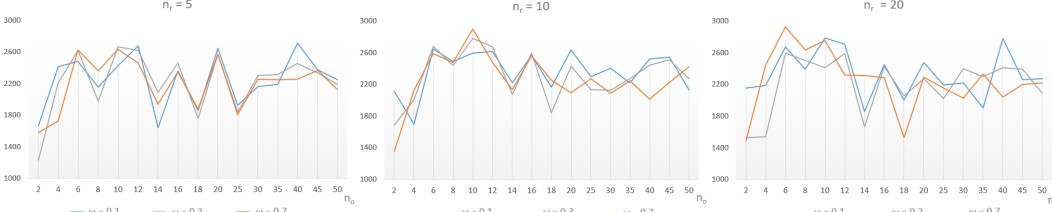

Figure 3: The episode reward for $n_r \in \{5, 10, 20\}$ as $n_o$ varies for DRC in Hopper.

Fig. 3 compares the performance of DRC as $n_o$, $n_r$, and $\omega$ vary in Hopper. There is a tradeoff of two forces: one is the reconstruction error turns zero when $n_o$ is a multiple of $n_r$; the other one is overfitting becomes worse as $n_o$ increases because samples are not infinite, which is discussed in Sec. 4.2.1. For small $n_r$ like 5 or 10, the performance of DRC reaches the best when $n_o = n_o$, but it is almost always worse when $n_o$ is a multiple of $n_r$. For large $n_r$ like 20, the performance of DRC decreases even when $n_o$ is smaller than $n_r$. Fig. 3 verifies our strategy of shooting $n_o = n_r$ of GDRC.

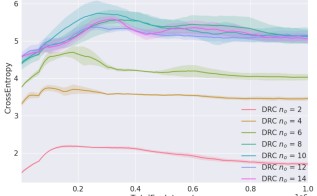

Figure 4: Cross entropy during the training for different values of $n_o$.

In Fig. 4, we study the empirical impact of $n_o$ on cross entropy over the course of training. We see that, indeed, cross-entropy increases rapidly for small $n_o$ and stops increasing when $n_o = n_r$, which proves the simulation results in Fig. 2 in real experiments. This suggests that identifying when cross entropy stops increasing can be a viable strategy for selecting $n_o$ as discussed in Section 4.2.1.

## 5.3   GCM PERTURBATIONS

**Discrete Control Tasks**   In Pendulum, the reward range $[-17, 0)$ is discretized into $n_r = 17$ bins. In CartPole, apart from $+1$ rewards, $-1$ rewards are introduced for perturbations by (Wang et al., 2020). GDRC is not needed in CartPole because rewards are discrete. As depicted in Fig. 5, DRC is always the best performing for all levels of noise. GDRC is the second strongest performer in Pendulum. Gaps between the methods become larger as the amount of noise increases. Despite the fact that

SR_W knows the confusion matrix a priori, it is not able to achieve a strong performance because it only uses the observed reward to condition to estimate the true reward and does not condition on the observed state and action. RE is a weaker performer because optimal-policy-unchanged does not hold.

**Mujoco Environments** It is not clear how to apply SR in these settings because the confusion matrix must be estimated. Therefore, only SR_W with a known confusion matrix is tested. Fig. 6 reveals DRC outperforming/tying with PPO, RE, and SR_W in 35/48 instances. Similarly, GDRC outperforms/ties in 33/48 cases against PPO and RE given the absence of pre-knowledge of perturbations, markedly surpassing the second-best performer at 12/48. Both DRC and GDRC demonstrate comprehensive robustness across varied environments, reward discretizations $n_r$, and noise ratios $\omega$.

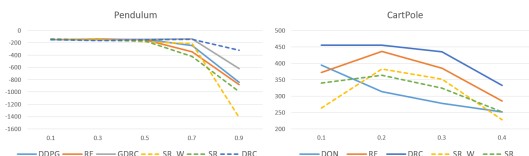

Figure 5: The results of Pendulum and CartPole under GCM perturbations. Solid line methods can be applied without any information. **DRC** and **GDRC** are our methods. The x-axis represents $\omega$.

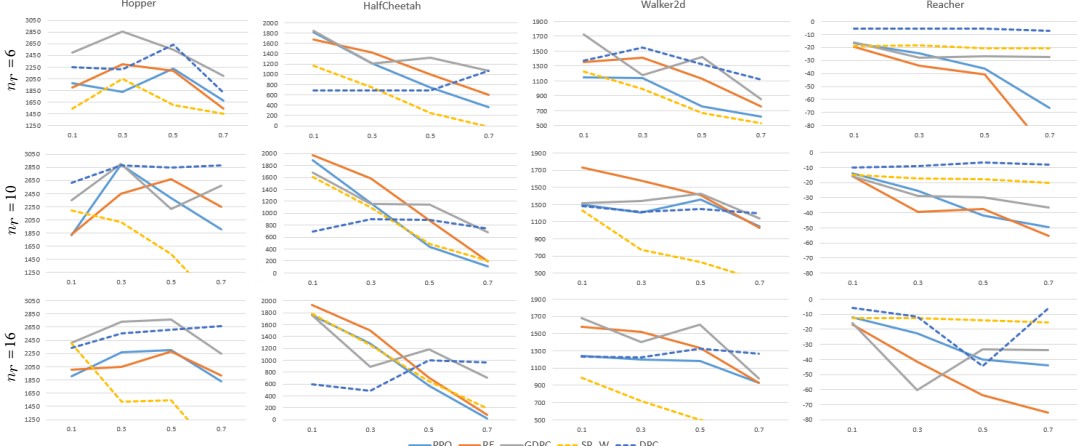

Figure 6: The results of Mujoco environments under GCM perturbations. Solid line methods can be applied without any information. **DRC** and **GDRC** are our methods. The $x$-axis represents $\omega$.

Performance generally diminishes with increasing $\omega$ across all environments, except for Hopper, which is because of the more exploration introduced by small perturbations as observed by (Wang et al., 2020). Benchmark performance in clean and totally perturbed (with $\omega = 1.0$ and $n_r \to \infty$) environments is provided for reference: 1844.2 and 1089.7 for Hopper, 1948.5 and -365.6 for HalfCheetah, 1286.6 and 756.7 for Walker, and -5.1 and -129.9 for Reacher. The agents in Hopper and Walker2d are encouraged to live longer because of the positive expectation of perturbed rewards because the perturbed and clean reward ranges are the same conditioned by (Wang et al., 2020).

One interesting phenomenon is that GDRC outperforms DRC in some settings, particularly in HalfCheetah. We perform further analysis in Fig. 10 in Appendix A, where we graph the reward critic training cross-entropy in the first row and true clean reward label distribution in the second row. We see that in DRC, the reward critic achieves very low cross-entropy and receives a highly imbalanced distribution of true rewards. We hypothesize that the reward critic has collapsed and is predicting the same value for all reward observations, essentially terminating the training prematurely as the few samples from the other class that are being received are not enough to shift the critic output. This collapse appears not to happen in GDRC—it is possible that the incorrect selection of $n_o$ leads to more random behavior initially, which allows escaping the region of the initially dominant reward class. In DRC in Hopper, we do not see signs of critic collapse. In the top left figure, we see a spike in the reward critic cross-entropy as the policy shifts and new rewards are observed, but such jumps are not always present (upper right). We discuss possible remedies to the collapse issue in Future Work.

## 5.4 Continuous Noise

We compare GRDC to PPO and RE on continuous noise distributions because SR_W, SR, and DRC cannot be applied without the concept of confusion matrices. In Fig. 7, GRDC wins/ties in 27/48 whereas RE wins/ties in 24/48 combinations. Even under non-GCM perturbations, GDRC has a small edge over RE. RE especially targets this kind of perturbation by making the stringent assumption that $\mathbb{E}(\tilde{r}) = \omega_0 \cdot r + \omega_1 (\omega_0 > 0, (\omega_0 \cdot r + \omega_1) \cdot r > 0)$.

We attribute GDRC's advantage to two factors. First, the continuous noise distributions are weakly mode-preserving in the sense that they have a peak in the observed reward distribution at true rewards. Second, we hypothesize that the distributional aspect of GDRC leads to more stable reward estimation.

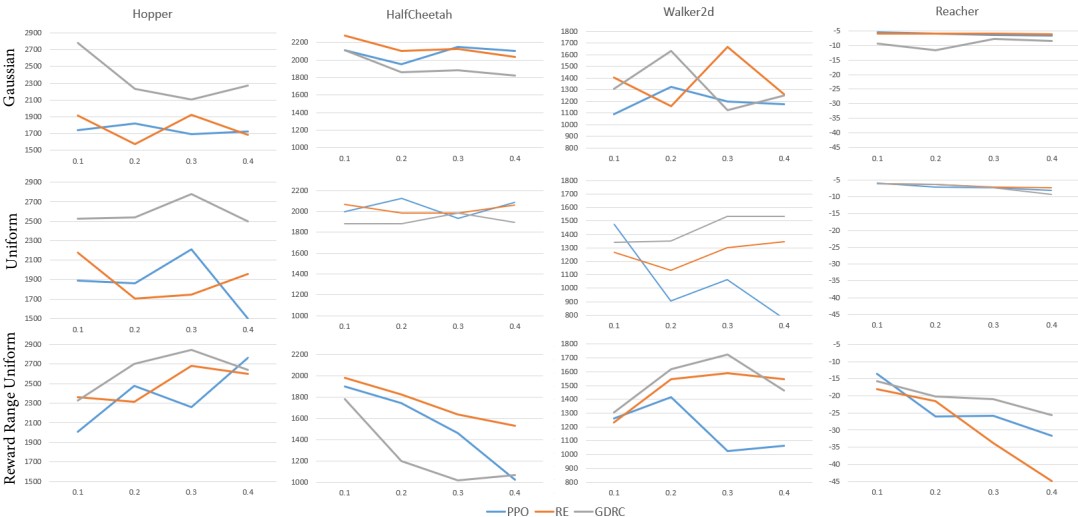

Figure 7: The results in Mujoco tasks under continuous perturbations. The $x$-axis represents $\omega$.

## 6 Conclusion

We study the impact of arbitrary mode-preserving reward perturbations in RL. We introduce a new definition of confusion matrix reward perturbations that generalizes past definitions. We propose a distributional reward critic method and analyze its behavior along key axes in the infinite sample case. We find that it achieves higher rewards than existing methods in a majority of domains and perturbations that we tested. It is indeed possible for a uniform reward estimation method to perform well across a large variety of reward perturbations.

**Future Work**  We find that environments where the reward distribution is highly skewed, such as HalfCheetah, are challenging for the reward critic methods due to sample class imbalance. Unlike in a classification setting where the true class labels are observable, it is not possible to directly perceive the true rewards. It is unclear whether it is possible to reweight the samples using state-action pairs, similar to Prioritized Experience Replay (PER) (Krishnamachari et al., 2019). Another approach to this problem would be to try to prevent the reward critic from converging on a single value, e.g., by adding an entropy bonus to its outputs.

Due to consecutive sample collections, a strong correlation exists among samples concerning states and actions, relaxing the issue of finite samples discussed in Section 5.2. Inspired by the concept of the replay buffer (Liu & Zou, 2018), originally aimed at decoupling sample correlation, we propose storing samples for reward critic fitting to further relax this assumption. More critically, the end goal is to achieve a balanced pace between two processes: the sample collection for reward critic fitting and the policy update, utilizing all available information during training. This approach is anticipated to alleviate the issue of imbalanced samples, ensuring the policy is updated before the collapse of the reward critic with the samples.

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

## A  IMPORTANT FIGURES

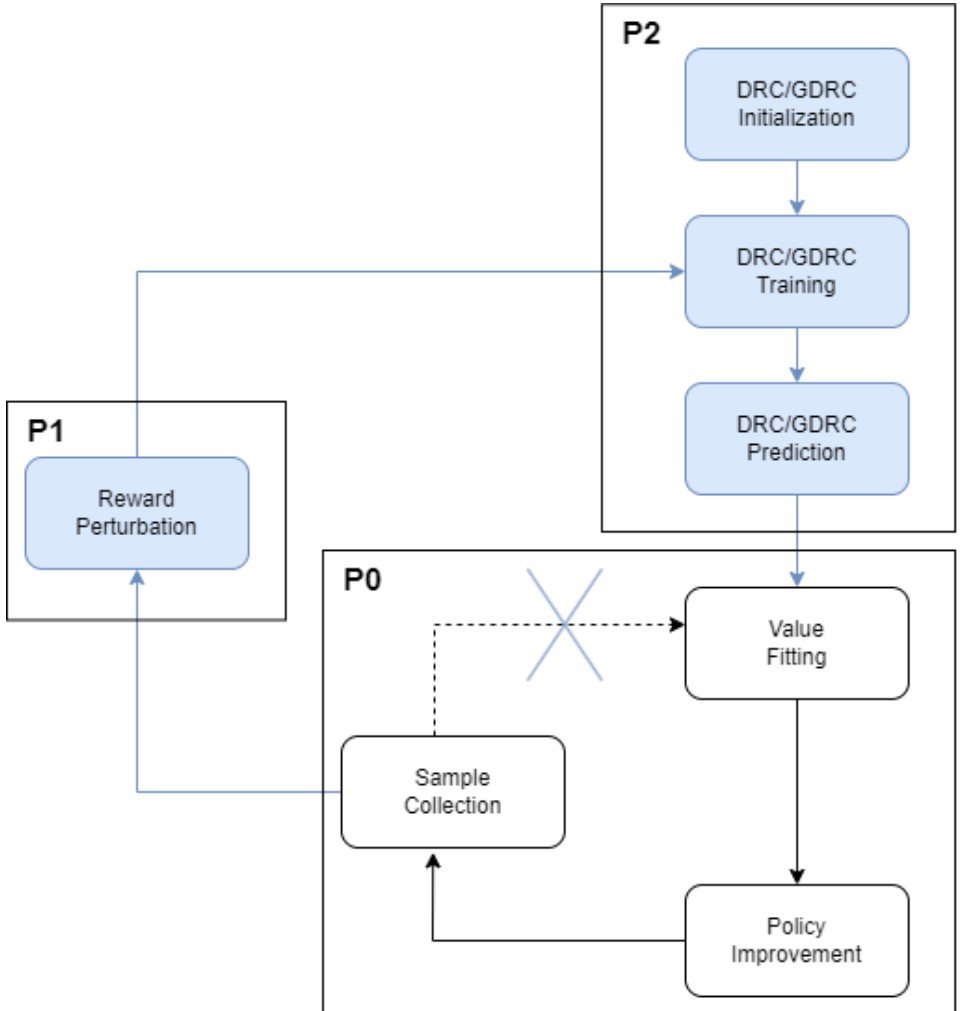

Figure 8: The flow chart to help understand the pipeline of the problem and how people resolve it. P0 represents the general pipeline of RL. P1 represents the perturbation applied, of which Fig. 9 is a concrete example to show how (generalized confusion matrix) GCM perturbation is introduced. P2 represents how people deal with the perturbation. Using our methods (DRC and GDRC) as examples, Alg. 1 and Alg. 2 are what we do in $P2$.

## B  PROOFS

### B.1  PROPOSITION 1

**Proposition 1.** *Consider a continuous perturbation model that for each reward $r \in [r_{\min}, r_{\max})$, it can be perturbed to $\bar{r} \in [r_{\min}, r_{\max})$. Our confusion matrix represents $\bar{r}$ with $\tilde{r}$ that satisfies $|\tilde{r} - \bar{r}| \leq \frac{r_{\max} - r_{\min}}{n_r}$.*

The Lipschitz constant $L$ of a continuous and bounded perturbation distribution is defined as Equation 1.

$$|f(x_2) - f(x_1)| \leq L \cdot |x_2 - x_1|, \ \forall x_1 \text{ and } x_2 \text{ within } [r_{min}, r_{max}) \tag{1}$$

As the probability within an interval stays unchanged while using a confusion matrix to model the continuous perturbation, the error at most is $L \cdot \frac{r_{max} - r_{min}}{N}$ as shown in Equation 2.

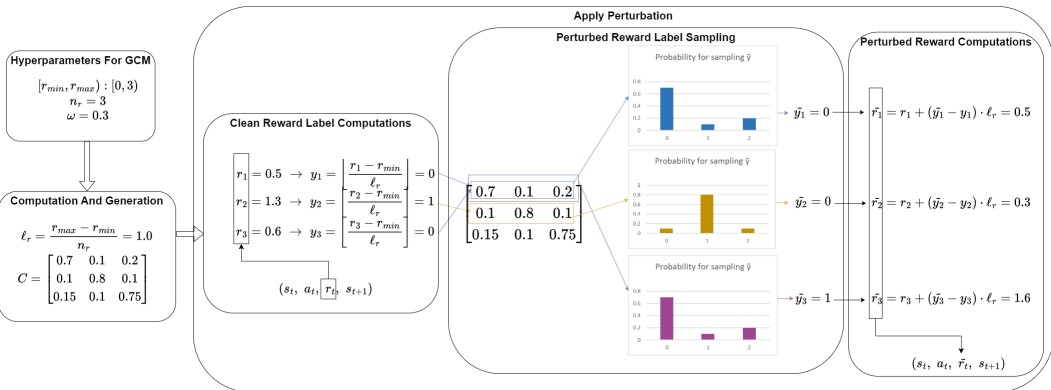

Figure 9: An example of the application of a GCM perturbation.

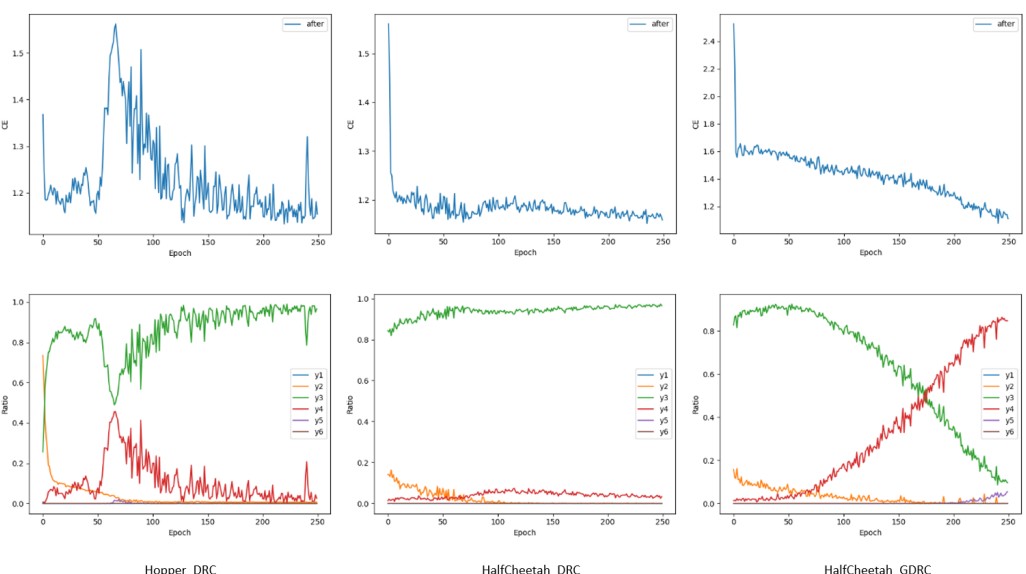

Figure 10: The results of the agents trained under the perturbation with $n_r = 6$ and $\omega = 0.1$. The two rows represent the cross-entropy of the reward critic prediction after the training and the (inaccessible) clean reward label ($y = \text{floor}\big((r - r_{\min})/\ell_r\big)$) distribution, and the three columns represent Hopper with DRC, HalfCheetah with DRC, and HalfCheetah with GDRC respectively.

$$|f(a_2) - f(a_1)| \leq L \cdot |a_2 - a_1| \leq L \cdot \left| a_1 + \frac{r_{max} - r_{min}}{N} - a_1 \right| = L \cdot \frac{r_{max} - r_{min}}{N}, \quad (2)$$

for $a_2$ and $a_1$ within the same interval.

## B.2 THEOREM 1

**Theorem 1.** *With a sufficiently expressive neural network and GCM perturbations, the prediction from DRC for a sample $(s_t, a_t)$ is $C(y_t, :)$ (the $y_t$-th row of $C$), which is the distribution of the perturbed reward label, as the number of samples from each state-action pair approaches infinity. Consequently, the correct reward can be exactly predicted if the perturbation is mode-preserving with known discretization.*

This proof assumes an abundance of samples with identical state-action pairs and a distributional reward critic powerful enough to generate any outputs. We define several notations. $\tilde{Y}(s, a) \in \mathbb{Z}$ represents the noisy reward labels of samples with identical $(s, a)$. We denote $\hat{R}_\theta = f_\theta(s, a) \in \mathbb{R}^K$, where $K = n_r$, as the predicted reward distribution with constraints: $\hat{R}_\theta[k] \geq 0$ and $\sum_{k=1}^{K} \hat{R}_\theta[k] = 1$. The probability of samples with varying noisy reward label $k$ is denoted by $p_k = P(\tilde{Y}(s, a) = k)$. $\tilde{y}$ is the one-hot representation of $\tilde{Y}$, and $\tilde{y}_k = \begin{bmatrix} 0 \\ \cdots \\ 1 \\ 0 \end{bmatrix}$ represents the sample with reward label $k$. The global loss function of all collected samples is presented in Equation 3:

$$\min_\theta \sum_t \mathcal{L}_{\text{CE}}\left(\tilde{y}(t), \hat{R}_\theta(t)\right) \quad (3)$$

where $\tilde{y}(t)$ and $\hat{R}_\theta(t)$ correspond to the actual expressions for $\tilde{y}$ and $\hat{R}_\theta$ respective to sample $(s_t, a_t)$. Given the power of the distributional reward critic and the abundance of samples for each state-action pair, the optimal solution of the local loss for a state-action pair decides the global one. Thus, we focus on the local loss function for identical state-action pairs $(s, a)$, neglecting subscript $t$, as in Equation 4:

$$\min_\theta \mathcal{L}_{\text{CE}}(\tilde{y}, \hat{R}_\theta) \quad (4)$$

The Cross-Entropy loss is utilized, and we group samples by their noisy reward labels as in Equation 5:

$$\min_\theta \sum_{k=1}^{K} p_k \cdot \mathcal{L}_{\text{CE}}(\tilde{y}_k, \hat{R}_\theta) \quad (5)$$

Referencing the definition of $y_k$, we expand the term $\mathcal{L}_{\text{CE}}(y_k, \hat{R}_\theta)$ as in Equation 6:

$$\mathcal{L}_{\text{CE}}(\tilde{y}_k, \hat{R}_\theta) = -\sum_j \tilde{y}_k[j] \cdot \log\left(\hat{R}_\theta[j]\right) = -\log\left(\hat{R}_\theta[k]\right) \quad (6)$$

Incorporating the final results of Equation 6 into Equation 5, we obtain Equation 7:

$$\min_\theta \sum_{k=1}^{K} -p_k \cdot \log\left(\hat{R}_\theta[k]\right) \quad (7)$$

With the definition of the Cross-Entropy loss, Equation 7 can be compressed into Equation 8:

$$\min_{\theta} \mathcal{L}_{\text{CE}}(p, \hat{R}_{\theta}), \ where \ p = \begin{bmatrix} p_1 \\ p_2 \\ .. \\ p_K \end{bmatrix} \tag{8}$$

Decomposing the cross-entropy in Equation 8 into entropy and KL-divergence results in Equation 9:

$$\mathcal{L}_{\text{CE}}(p, \hat{R}_{\theta}) = H(p, \hat{R}_{\theta}) = H(p) + D_{KL}(p \,||\, \hat{R}_{\theta}) \tag{9}$$

Here, $H(p)$ represents the entropy of $p$, unaffected by $\theta$, and $D_{KL}(p \,||\, \hat{R}_{\theta})$ denotes the KL-divergence between $p$ and $\hat{R}_{\theta}$. $D_{KL}(p \,||\, \hat{R}_{\theta})$ only reaches its minimum of zero if $\hat{R}_{\theta} = p$ is satisfied. Consequently, the optimal solution is $\hat{R}_{\theta} = p$, signifying that the post-training reward critic output is the distribution of the perturbed reward labels.

### B.3 THEOREM 2

**Theorem 2.** *Consider the infinite sample limit. When $n_o \leq n_r$, $\min_{p_o} H(p_{r \to o}, p_o)$ is non-decreasing in $n_o$. Moreover, if $P(C(y) = y') > 0$ for $\forall y, y' \in Y$, $\min_{p_o} H(p_{r \to o}, p_o)$ is an increasing function of $n_o$. When $n_o \geq n_r$, $\min_{p_o} H(p_{r \to o}, p_o) = H(p_{r \to r})$, the entropy of $p_r$.*

With the optimal solution of the loss function we prove above, $\min_{p_o} H(p_{r \to o}, p_o)$ can be simplified as $H(p_{r \to o})$. For simplicity in this proof, we replace $n_r$ and $n_o$ with $m$ and $k$ respectively. The central question here is to investigate the relationship between $H(p) = -\sum_k^K p_k \log p_k$ and $K$, where the original noise distribution is characterized by $M$ intervals.

#### B.3.1 PROBLEM MODELING

For modeling the noisy rewards, the rewards are assumed to lie within the range $[0, M)$ with $M$ intervals where each interval length is 1. For example, if $r \in [0, 1)$, the perturbation is applied as per $\tilde{r} = r + m$ for $m \in [0, M)$, where $m$ is an integer. The discrete probability distribution of $\tilde{r}$ is denoted as $q$, where $q_m = P(\tilde{r} = r + m)$.

For modeling the training, we are assumed to know the reward range $[0, M)$, but we are not informed about the number of discretization $M$. Instead, we fit the reward range with a discretized distribution having $K$ intervals. Given a noisy reward $\tilde{r}$, we compute $\tilde{y} = \lfloor \tilde{r}/\frac{M}{K} \rfloor$ correspondingly. The $p$ is defined in Equation 10, which captures the mapping from $q$ to $p$.

$$p_k = P[\tilde{y} = k] = P[k \cdot \frac{M}{K} \leq \tilde{r} \leq (k+1) \cdot \frac{M}{K}], k = 0, 1, \ldots, K-1 \tag{10}$$

#### B.3.2 WHEN $\mathbf{K \leq M}$

The expression for $p_k$ in terms of $q_m$ is given in Equation 11:

$$p_k = \sum_{m:k\frac{M}{K} \leq r+m \leq (k+1)\frac{M}{K}} q_m \tag{11}$$

It is known that $\forall q_0, q_1 \geq 0, q_0 + q_1 \leq 1$, it follows that $-(q_0 + q_1)\log(q_0 + q_1) = -(q_0 \log(q_0 + q_1) + q_1 \log(q_0 + q_1)) \leq -(q_0 \log q_0 + q_1 \log q_1)$. Equality only holds if $q_0 = 0$ or $q_1 = 0$. However, this is not possible in our confusion matrix cases because each cell has a value greater than 0.

As Fig. 11a illustrates, more $q_m$ get combined together for smaller $K$, resulting in a smaller $H(p) = -\sum_k^K p_k \log p_k$. specifically, the number of $q_m$ that are combined equals $M - K$. Therefore, $H(p)$ is non-decreasing concerning $K$, satisfying the first part of Theorem 2.

### B.3.3 WHEN $\mathbf{K} > \mathbf{M}$

As illustrated in Fig. 11b, there are dimensions of $p$ with zero probabilities because $K > M$, where the number of those dimensions equals $K - M$. Hence, $H(p) = -\sum_k^K p_k \log p_k = -\sum_m^M q_m \log q_m$ remains constant with respect to $K$, corresponding to the second half of Theorem 2.

Moreover, this theorem can also be validated empirically through simulations under unlimited samples, as presented in Fig. 2.

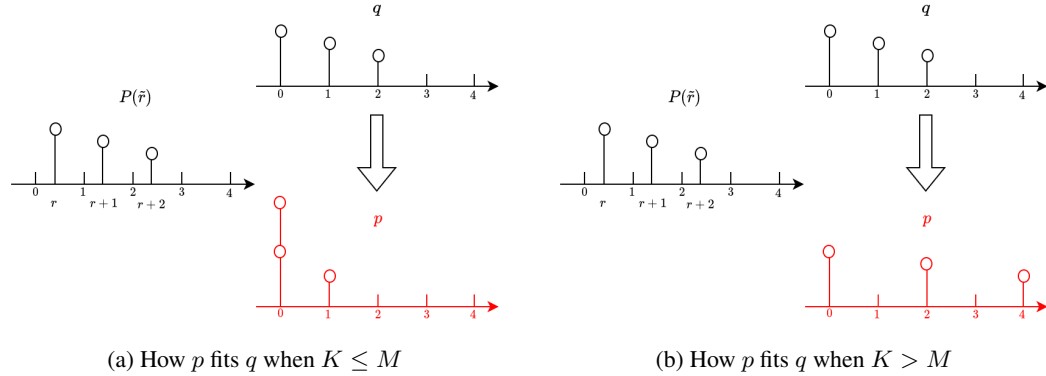

(a) How $p$ fits $q$ when $K \leq M$        (b) How $p$ fits $q$ when $K > M$

Figure 11: Illustration of the cross entropy as for different $K$.

## C  MORE ANALYSIS OF THE RECONSTRUCTION ERROR

The theoretical discussion on the reconstruction error, ERROR$_r$, is detailed in Section 4.2.1, and empirical evidence from simulation experiments is depicted in Fig.1. We want to extract several key observations from Fig. 1 and correlate them with the theoretical discussions, as follows:

- In the cases presented in Fig. 1, $n_r$ is set to 10. It is apparent that the reconstruction error converges to zero for $n_o = 10, 20, 30, 40, 50$ with a standard deviation of zero, suggesting the noisy rewards can be reverted to the clean rewards exactly when $n_o$ is a multiple of $n_r$ under the premise of infinite samples.
- The value of ERROR$_r$ is lower when $n_o < n_r$ in comparison to when $n_o > n_r$. This suggests that choosing a larger value for $n_o$, without knowledge of $n_r$, may be preferable. However, an overly large value for $n_o$ could lead to overfitting in practical cases, as detailed in Section 4.2.1.
- In conjunction with Fig. 2, it is evident that cross-entropy provides a clear indication of an optimal point when $n_o = n_r$, In this situation, concerns about overfitting are minimized, and we concurrently achieve a reconstruction error of zero.

## D  PSEUDOCODE OF GDRC

Alg. 2 presents the pseudocode for implementing GDRC with the known range, as discussed in Section 4.2.1.

## E  EXPERIMENTAL HYPERPARAMETERS

In essence, our experimentation is conducted via both the Spinning Up and Stable Baselines3 frameworks, for which we detail their important hyperparameter configurations separately. The source codes associated with this work will be released for public access upon acceptance.

**Spinning Up**  The environments of Hopper-v3, HalfCheetah-v3, Walker2d-v3, and Reacher-v2 are trained using PPO-associated algorithms. Adam optimizers are employed for all neural network training processes. A total of 4,000 steps per epoch is configured and the agents are trained across 250 epochs, resulting in an aggregate of 1,000,000 interactions. The maximum episodic length is

---

**Algorithm 2** General Distributional Reward Critic (GDRC)

---

1: **procedure** GDRC INITIALIZATION
2:     **Notations:** $[n_1, n_2, ..., n_k]$: the quantities of outputs from the set of DRC under consideration
3:     Initialize $k$ reward critics, denoted as $\mathrm{DRC}_n$, with output quantities $n_1, n_2, \ldots, n_k$.
4:     **Input:** $s \times a$
5:     **Output:** $[0, 1]^n$, $n \in [n_1, n_2, \ldots, n_k]$

6: **procedure** GDRC TRAINING
7:     **Notations:** $\tilde{y}_n$: noisy reward label regarding $n$; Stream_Buf: the streaming buffer to store the rewards of the collected samples; $r_{\mathrm{emin}}$: the $5\%$ percentile of the collected rewards; $r_{\mathrm{emax}}$: the $95\%$ percentile of the collected rewards; $\ell_n$: the length of each reward interval regarding $n$, $\ell_n = (r_{\mathrm{emax}} - r_{\mathrm{emin}})/n$
8:     **Input:** samples $(s, a, \tilde{r})s$ collected in an epoch
9:     **Objective:** parallel training of the reward critics and execution of voting
10:     Time the voted numbers by a discount factor.
11:     Store $(r)s$ into Stream_Buf and set $r_{\mathrm{emin}}$ and $r_{\mathrm{emax}}$ to $5\%$ and $95\%$ percentiles respectively.
12:     **for** $n$ **in** $[n_1, n_2, ..., n_k]$ **do**
13:         Calculate $\ell_n$: $\ell_n = \big( (r_{\mathrm{emax}} - r_{\mathrm{emin}})/n \big)$
14:         Convert $\tilde{r}$ to $\tilde{y}_n$ by applying: $\tilde{y}_n = floor\big( (\tilde{r} - r_{\mathrm{emin}})/\ell_n \big)$
15:         **while** training iterations threshold not reached **do**
16:             Train $\mathrm{DRC}_n$ using inputs $(s, a)$ and labels $(\tilde{y}_n)$
17:     **for** $n$ **in** $[n_1, n_2, ..., n_k]$ **do**
18:         Perform prediction using $\mathrm{DRC}_n$ and derive $\hat{R}_{\theta n}$
19:         Compute cross entropy $H_n = H(\tilde{y}_n, \hat{R}_{\theta n})$
20:         Compute $dH_n = H_n - H_{n\prime}$, where $n\prime$ denotes the previous $n$
21:         **if** $dH_n > dH_{n\prime}$ **then**
22:             Cast vote for $n\prime$
23:     Select $\mathrm{DRC}_n$ that received the maximum votes, with $n$ denoting the number of discretization

24: **procedure** GDRC PREDICTING$(s_t, a_t, \tilde{r}_t)$
25:     **Notations:** $r_{\mathrm{emin}}$: the $5\%$ percentile of the collected rewards; $r_{\mathrm{emax}}$: the $95\%$ percentile of the collected rewards; reward critic $\hat{R}_\theta$
26:     **Input:** $(s_t, a_t, \tilde{r}_t)$
27:     **Output:** $(s_t, a_t, \hat{r}_t)$
28:     Select $\mathrm{DRC}_n$ that received the maximum votes, with $n$ denoting the number of discretization
29:     Compute the length of each reward interval $\ell_n = (r_{\mathrm{emax}} - r_{\mathrm{emin}})/n$
30:     Associate each sample with a discrete label $\tilde{y}_t = \mathrm{floor}\big( (\tilde{r}_t - r_{\mathrm{emin}})/\ell_n \big)$
31:     Input $(s_t, a_t)$ to $\mathrm{DRC}_n$, and obtain $\hat{R}_\theta(s_t, a_t)$.
32:     Determine the predicted reward label $\hat{y}_t$: $\hat{y}_t = \mathrm{argmax}\, \hat{R}_\theta(s_t, a_t)$
33:     Compute the predicted reward value $\hat{r}_t$: $\hat{r}_t = \tilde{r}_t + (\hat{y}_t - \tilde{y}_t) \cdot \ell_n$

---

designated as 1,000. Learning rates of 3e-4 and 1e-3 are applied to the policy and the value function respectively, with training conducted over 80 iterations per update. The clipping ratio, used in the policy objective, is set at 0.2. The GAE-Lambda parameter is set at 0.97. For methods incorporating the surrogate reward (SR) method, the reward critic has a learning rate of 1e-3 and trains over 80 iterations. For those involving a Distributional Reward Critic (DRC), the reward critic adopts a learning rate of 1e-3 and trains over 40 iterations per update.

Pendulum-v1 is trained via DDPG-associated algorithms. Adam optimizers are employed for all neural network training processes. Configured for 200 steps per epoch, the agents are trained over 750 epochs, resulting in a total of 150,000 interactions. The maximum episodic length is set at 200. Learning rates of 1e-3 are assigned to both the policy and the Q-function, with updates occurring every 200 steps. The size of the replay buffer is 10e5 and the batch size is set at 200. When a surrogate reward (SR) method is employed, the hyperparameters are aligned with (Wang et al., 2020). The hyperparameter settings for DRC-related methods remain unchanged.

**Stable Baselines3** The hyperparameters utilized in Stable Baselines3 are derived from RL Baselines3 Zoo. CartPole-v1 is trained with DQN-associated algorithms. Adam optimizers are utilized for all neural network training. The agent undergoes a total of 50,000 steps in training. The learning rate for the optimizer is set at 0.0023. The fraction of the total training period during which the exploration rate diminishes is set at 0.16, with the exploration probability dropping from 1.0 to 0.04. The buffer size is sufficiently large to accommodate all samples. When SR or DRC-related methods are applied, the hyperparameters remain unchanged as we discuss previously.

## F  EXPERIMENT RESULT TABLES

Table 2: Experiement Results in Pendulum Under GCM perturbations. Bolded methods are our methods. Blue methods can be applied without any information about the perturbations.

| $\omega$ | Algs | Pendulum |
|---|---|---|
| $\omega = 0.1$ | DDPG | -149.7 +/- 6.1 |
| | RE | -147.3 +/- 5.8 |
| | **GDRC** | -149.8 +/- 5.8 |
| | SR_W | -138.4 +/- 6.3 |
| | SR | -135.2 +/- 5.5 |
| | **DRC** | -151.1 +/- 6.2 |
| $\omega = 0.3$ | DDPG | -149.9 +/- 6.5 |
| | RE | -136.0 +/- 6.1 |
| | **GDRC** | -148.2 +/- 6.0 |
| | SR_W | -142.0 +/- 5.3 |
| | SR | -161.4 +/- 6.4 |
| | **DRC** | -162.6 +/- 6.5 |
| $\omega = 0.5$ | DDPG | -144.3 +/- 6.0 |
| | RE | -149.9 +/- 6.6 |
| | **GDRC** | -144.5 +/- 6.3 |
| | SR_W | -184.4 +/- 15.1 |
| | SR | -167.3 +/- 6.4 |
| | **DRC** | -148.1 +/- 5.7 |
| $\omega = 0.7$ | DDPG | -244.9 +/- 14.0 |
| | RE | -347.9 +/- 28.9 |
| | **GDRC** | -137.5 +/- 6.0 |
| | SR_W | -215.7 +/- 12.6 |
| | SR | -422.3 +/- 21.1 |
| | **DRC** | -144.2 +/- 6.0 |
| $\omega = 0.9$ | DDPG | -843.7 +/- 31.9 |
| | RE | -879.3 +/- 36.2 |
| | **GDRC** | -621.7 +/- 35.5 |
| | SR_W | -1413.0 +/- 21.7 |
| | SR | -992.9 +/- 31.7 |
| | **DRC** | -320.9 +/- 30.9 |

Table 3: Experiement Results in CartPole Under GCM perturbations. Bolded methods are our methods. Blue methods can be applied without any information about the perturbations.

| $\omega$ | Algs | CartPole |
|---|---|---|
| | DQN | 394 +/- 13 |
| | RE | 372 +/- 13 |
| $\omega = 0.1$ | **DRC** | 455 +/- 9 |
| | SR_W | 264 +/- 13 |
| | SR | 340 +/- 13 |
| | DQN | 313 +/- 14 |
| | RE | 436 +/- 9 |
| $\omega = 0.2$ | **DRC** | 455 +/- 9 |
| | SR_W | 382 +/- 11 |
| | SR | 364 +/- 12 |
| | DQN | 278 +/- 13 |
| | RE | 385 +/- 11 |
| $\omega = 0.3$ | **DRC** | 435 +/- 13 |
| | SR_W | 352 +/- 13 |
| | SR | 324 +/- 14 |
| | DQN | 252 +/- 12 |
| | RE | 285 +/- 13 |
| $\omega = 0.4$ | **DRC** | 332 +/- 13 |
| | SR_W | 228 +/- 13 |
| | SR | 252 +/- 14 |

Table 4: Experiement Results in Mujoco tasks Under GCM perturbations. Bolded methods are our methods. Blue methods can be applied without any information about the perturbations.

| $n_r$ | $\omega$ | Algs | Hopper | HalfCheetah | Walker2d | Reacher |
|---|---|---|---|---|---|---|
| $n_r = 6$ | $\omega = 0.1$ | PPO | 1968.9 +/- 46.6 | 1809.6 +/- 45.0 | 1143.3 +/- 24.3 | -16.7 +/- 0.5 |
| | | RE | 1904.1 +/- 43.3 | 1674.5 +/- 44.9 | 1351.3 +/- 35.7 | -19.6 +/- 0.5 |
| | | GDRC | 2495.1 +/- 35.3 | 1840.8 +/- 54.0 | 1722.0 +/- 35.6 | -16.3 +/- 0.3 |
| | | SR_W | 1542.4 +/- 59.7 | 1172.7 +/- 64.0 | 1226.1 +/- 29.1 | -19.1 +/- 0.3 |
| | | DRC | 2249.7 +/- 60.6 | 679.4 +/- 4.4 | 1368.1 +/- 35.0 | -5.5 +/- 0.1 |
| | $\omega = 0.3$ | PPO | 1827.8 +/- 64.0 | 1212.5 +/- 45.1 | 1141.0 +/- 40.8 | -24.8 +/- 1.0 |
| | | RE | 2294.7 +/- 49.6 | 1424.7 +/- 39.8 | 1410.6 +/- 29.9 | -34.3 +/- 1.3 |
| | | GDRC | 2855.8 +/- 22.6 | 1201.6 +/- 44.5 | 1178.3 +/- 43.6 | -27.8 +/- 0.8 |
| | | SR_W | 2053.9 +/- 46.4 | 748.2 +/- 66.4 | 995.5 +/- 17.8 | -18.4 +/- 0.2 |
| | | DRC | 2208.4 +/- 66.9 | 689.1 +/- 4.0 | 1549.6 +/- 33.9 | -5.5 +/- 0.1 |
| | $\omega = 0.5$ | PPO | 2225.8 +/- 50.1 | 740.0 +/- 28.8 | 754.2 +/- 21.0 | -36.4 +/- 2.4 |
| | | RE | 2180.3 +/- 47.7 | 990.8 +/- 40.7 | 1131.8 +/- 36.5 | -40.6 +/- 2.2 |
| | | GDRC | 2541.3 +/- 37.5 | 1321.4 +/- 45.7 | 1419.6 +/- 28.4 | -26.6 +/- 0.9 |
| | | SR_W | 1600.6 +/- 60.9 | 252.7 +/- 45.0 | 673.7 +/- 15.4 | -20.5 +/- 0.3 |
| | | DRC | 2633.0 +/- 10.8 | 692.2 +/- 3.5 | 1322.7 +/- 44.5 | -5.6 +/- 0.1 |
| | $\omega = 0.7$ | PPO | 1672.9 +/- 47.9 | 365.2 +/- 28.6 | 621.1 +/- 17.2 | -66.6 +/- 3.5 |
| | | RE | 1536.1 +/- 50.9 | 595.5 +/- 14.5 | 755.0 +/- 34.2 | -100.9 +/- 6.6 |
| | | GDRC | 2100.3 +/- 68.5 | 1064.1 +/- 61.0 | 856.9 +/- 36.1 | -27.6 +/- 1.1 |
| | | SR_W | 1455.1 +/- 45.6 | -13.6 +/- 10.1 | 536.6 +/- 13.3 | -20.7 +/- 0.3 |
| | | DRC | 1814.4 +/- 65.1 | 1069.2 +/- 52.9 | 1120.0 +/- 51.0 | -7.1 +/- 0.2 |
| $n_r = 10$ | $\omega = 0.1$ | PPO | 1814.6 +/- 40.1 | 1882.8 +/- 44.6 | 1294.3 +/- 29.2 | -13.9 +/- 0.2 |
| | | RE | 1829.1 +/- 30.6 | 1977.1 +/- 56.7 | 1729.9 +/- 34.9 | -15.6 +/- 0.3 |
| | | GDRC | 2353.2 +/- 36.2 | 1685.5 +/- 47.8 | 1311.2 +/- 35.6 | -15.7 +/- 0.2 |
| | | SR_W | 2195.9 +/- 40.9 | 1605.7 +/- 42.7 | 1228.7 +/- 33.4 | -15.0 +/- 0.2 |
| | | DRC | 2614.4 +/- 41.1 | 692.3 +/- 2.2 | 1278.3 +/- 11.7 | -9.8 +/- 0.2 |
| | $\omega = 0.3$ | PPO | 2895.3 +/- 32.6 | 1173.6 +/- 54.7 | 1201.2 +/- 27.0 | -25.3 +/- 0.9 |
| | | RE | 2448.8 +/- 30.5 | 1587.6 +/- 54.5 | 1573.5 +/- 26.5 | -39.6 +/- 1.7 |
| | | GDRC | 2899.2 +/- 11.2 | 1153.9 +/- 24.8 | 1342.7 +/- 54.8 | -28.6 +/- 1.0 |
| | | SR_W | 2017.1 +/- 70.8 | 1095.8 +/- 33.1 | 772.2 +/- 33.0 | -17.2 +/- 0.3 |
| | | DRC | 2883.3 +/- 4.3 | 898.0 +/- 27.5 | 1211.0 +/- 23.6 | -9.0 +/- 0.2 |
| | $\omega = 0.5$ | PPO | 2378.5 +/- 55.7 | 434.1 +/- 36.4 | 1355.3 +/- 28.1 | -41.6 +/- 1.8 |
| | | RE | 2670.9 +/- 36.2 | 871.3 +/- 67.9 | 1410.8 +/- 23.3 | -37.7 +/- 1.9 |
| | | GDRC | 2219.3 +/- 56.3 | 1139.4 +/- 27.6 | 1427.4 +/- 40.7 | -29.6 +/- 0.6 |
| | | SR_W | 1534.0 +/- 75.5 | 479.7 +/- 19.9 | 632.9 +/- 26.0 | -17.8 +/- 0.2 |
| | | DRC | 2849.5 +/- 9.6 | 886.6 +/- 27.5 | 1247.3 +/- 19.1 | -6.5 +/- 0.1 |
| | $\omega = 0.7$ | PPO | 1905.1 +/- 56.4 | 104.0 +/- 33.8 | 1041.2 +/- 22.0 | -49.6 +/- 2.2 |
| | | RE | 2245.6 +/- 39.7 | 189.1 +/- 35.9 | 1028.3 +/- 20.4 | -55.4 +/- 2.4 |
| | | GDRC | 2572.0 +/- 36.6 | 678.8 +/- 61.1 | 1137.3 +/- 32.2 | -36.5 +/- 1.4 |
| | | SR_W | 728.0 +/- 27.5 | 210.0 +/- 14.8 | 418.5 +/- 5.9 | -20.1 +/- 0.5 |
| | | DRC | 2879.4 +/- 4.6 | 739.6 +/- 13.0 | 1199.2 +/- 35.5 | -8.1 +/- 0.6 |
| $n_r = 16$ | $\omega = 0.1$ | PPO | 1901.1 +/- 36.9 | 1760.9 +/- 63.4 | 1239.8 +/- 34.6 | -12.3 +/- 0.2 |
| | | RE | 2006.6 +/- 36.9 | 1928.1 +/- 57.5 | 1580.5 +/- 27.4 | -17.0 +/- 0.3 |
| | | GDRC | 2416.8 +/- 40.6 | 1759.4 +/- 44.4 | 1678.1 +/- 37.6 | -16.0 +/- 0.2 |
| | | SR_W | 2397.3 +/- 45.7 | 1781.3 +/- 52.2 | 989.4 +/- 35.8 | -12.6 +/- 0.1 |
| | | DRC | 2332.1 +/- 34.4 | 601.4 +/- 45.5 | 1235.6 +/- 29.9 | -5.9 +/- 0.1 |
| | $\omega = 0.3$ | PPO | 2266.0 +/- 55.5 | 1282.0 +/- 52.1 | 1195.7 +/- 22.9 | -22.7 +/- 1.0 |
| | | RE | 2050.2 +/- 53.1 | 1504.6 +/- 54.7 | 1520.7 +/- 21.3 | -41.7 +/- 1.4 |
| | | GDRC | 2732.5 +/- 10.8 | 893.1 +/- 28.1 | 1399.1 +/- 40.4 | -60.3 +/- 4.9 |
| | | SR_W | 1523.9 +/- 34.2 | 1258.9 +/- 54.3 | 718.8 +/- 23.2 | -12.5 +/- 0.1 |
| | | DRC | 2554.0 +/- 22.1 | 486.3 +/- 43.5 | 1224.3 +/- 29.8 | -11.5 +/- 1.0 |
| | $\omega = 0.5$ | PPO | 2303.3 +/- 55.0 | 567.4 +/- 24.9 | 1182.5 +/- 31.1 | -40.3 +/- 1.6 |
| | | RE | 2274.5 +/- 46.8 | 702.3 +/- 48.8 | 1337.1 +/- 28.6 | -63.6 +/- 2.6 |
| | | GDRC | 2765.0 +/- 18.4 | 1182.2 +/- 25.6 | 1606.3 +/- 23.2 | -33.4 +/- 1.1 |
| | | SR_W | 1542.6 +/- 56.0 | 645.8 +/- 28.0 | 497.0 +/- 16.8 | -14.1 +/- 0.1 |
| | | DRC | 2612.6 +/- 21.9 | 1004.5 +/- 47.8 | 1323.0 +/- 33.5 | -44.2 +/- 5.6 |
| | $\omega = 0.7$ | PPO | 1824.4 +/- 40.8 | 18.2 +/- 16.3 | 926.1 +/- 14.0 | -43.7 +/- 2.2 |
| | | RE | 1919.6 +/- 48.5 | 82.8 +/- 24.4 | 931.2 +/- 14.9 | -75.1 +/- 3.0 |
| | | GDRC | 2247.6 +/- 56.0 | 704.8 +/- 36.8 | 977.8 +/- 28.9 | -33.8 +/- 0.9 |
| | | SR_W | 663.2 +/- 23.6 | 196.2 +/- 12.8 | 371.0 +/- 3.0 | -15.4 +/- 0.2 |
| | | DRC | 2666.8 +/- 39.3 | 962.5 +/- 47.0 | 1269.4 +/- 22.8 | -6.1 +/- 0.2 |

Table 5: Experiement Results in Mujoco tasks Under continuous perturbations. Bolded methods are our methods.

| Perturbation | $\sigma/\omega$ | Algs | Hopper | HalfCheetah | Walker2d | Reacher |
|---|---|---|---|---|---|---|
| Gaussian | $\sigma = 0.1$ | PPO | 1741.7 +/- 40.9 | 2111.2 +/- 46.8 | 1092.4 +/- 30.1 | -5.6 +/- 0.1 |
| | | RE | 1912.7 +/- 38.6 | 2280.3 +/- 54.2 | 1407.4 +/- 55.0 | -5.9 +/- 0.1 |
| | | **GDRC** | 2779.5 +/- 26.4 | 2111.5 +/- 63.8 | 1308.6 +/- 32.3 | -9.4 +/- 0.1 |
| | $\sigma = 0.2$ | PPO | 1823.2 +/- 36.6 | 1953.3 +/- 48.0 | 1326.0 +/- 39.5 | -6.0 +/- 0.1 |
| | | RE | 1575.6 +/- 22.0 | 2105.7 +/- 55.2 | 1157.9 +/- 30.8 | -6.0 +/- 0.0 |
| | | **GDRC** | 2234.6 +/- 42.7 | 1862.7 +/- 55.6 | 1635.8 +/- 44.4 | -11.7 +/- 0.1 |
| | $\sigma = 0.3$ | PPO | 1691.6 +/- 30.7 | 2147.3 +/- 51.3 | 1200.9 +/- 30.2 | -6.4 +/- 0.1 |
| | | RE | 1927.9 +/- 34.0 | 2124.2 +/- 49.2 | 1666.0 +/- 38.8 | -6.0 +/- 0.1 |
| | | **GDRC** | 2107.8 +/- 39.8 | 1883.3 +/- 57.2 | 1124.4 +/- 30.3 | -7.8 +/- 0.1 |
| | $\sigma = 0.4$ | PPO | 1728.4 +/- 43.3 | 2103.2 +/- 49.1 | 1173.2 +/- 39.3 | -6.6 +/- 0.1 |
| | | RE | 1682.5 +/- 28.4 | 2034.3 +/- 59.4 | 1264.5 +/- 35.1 | -6.1 +/- 0.1 |
| | | **GDRC** | 2270.5 +/- 43.9 | 1818.7 +/- 50.4 | 1252.6 +/- 25.1 | -8.5 +/- 0.1 |
| Uniform | $\omega = 0.1$ | PPO | 1889.6 +/- 39.7 | 1995.0 +/- 55.4 | 1475.9 +/- 39.5 | -5.9 +/- 0.1 |
| | | RE | 2176.6 +/- 36.3 | 2067.2 +/- 57.5 | 1269.0 +/- 29.5 | -6.1 +/- 0.1 |
| | | **GDRC** | 2527.7 +/- 27.7 | 1884.9 +/- 52.5 | 1342.6 +/- 42.0 | -6.1 +/- 0.1 |
| | $\omega = 0.2$ | PPO | 1863.2 +/- 43.0 | 2127.7 +/- 48.6 | 905.3 +/- 30.6 | -7.0 +/- 0.1 |
| | | RE | 1703.6 +/- 30.7 | 1987.5 +/- 50.7 | 1134.1 +/- 33.4 | -6.2 +/- 0.1 |
| | | **GDRC** | 2541.4 +/- 44.7 | 1883.7 +/- 47.1 | 1348.9 +/- 54.3 | -6.3 +/- 0.0 |
| | $\omega = 0.3$ | PPO | 2208.9 +/- 37.3 | 1934.5 +/- 57.3 | 1065.0 +/- 39.7 | -7.3 +/- 0.1 |
| | | RE | 1743.2 +/- 43.8 | 1986.6 +/- 61.7 | 1300.8 +/- 21.9 | -7.0 +/- 0.1 |
| | | **GDRC** | 2776.8 +/- 26.7 | 1986.6 +/- 54.0 | 1533.0 +/- 39.1 | -7.2 +/- 0.1 |
| | $\omega = 0.4$ | PPO | 1500.6 +/- 37.6 | 2085.7 +/- 49.7 | 771.5 +/- 41.3 | -8.1 +/- 0.1 |
| | | RE | 1957.7 +/- 24.5 | 2063.2 +/- 54.4 | 1347.3 +/- 37.4 | -7.3 +/- 0.1 |
| | | **GDRC** | 2498.0 +/- 26.9 | 1892.6 +/- 49.7 | 1534.3 +/- 43.5 | -9.3 +/- 0.1 |
| Reward Range Uniform | $\omega = 0.1$ | PPO | 2009.4 +/- 34.9 | 1898.0 +/- 54.1 | 1260.2 +/- 35.4 | -13.6 +/- 0.3 |
| | | RE | 2364.1 +/- 31.7 | 1977.2 +/- 54.3 | 1230.5 +/- 28.3 | -18.0 +/- 0.6 |
| | | **GDRC** | 2327.4 +/- 39.5 | 1778.3 +/- 46.8 | 1303.8 +/- 41.7 | -15.8 +/- 0.3 |
| | $\omega = 0.2$ | PPO | 2475.0 +/- 43.4 | 1742.9 +/- 60.7 | 1418.2 +/- 36.1 | -26.1 +/- 0.7 |
| | | RE | 2316.7 +/- 30.0 | 1822.5 +/- 62.1 | 1544.6 +/- 29.4 | -21.5 +/- 0.5 |
| | | **GDRC** | 2701.8 +/- 29.1 | 1200.9 +/- 30.4 | 1618.2 +/- 33.0 | -20.1 +/- 0.6 |
| | $\omega = 0.3$ | PPO | 2260.1 +/- 24.4 | 1463.6 +/- 59.0 | 1024.5 +/- 23.7 | -25.8 +/- 0.7 |
| | | RE | 2679.0 +/- 22.4 | 1633.8 +/- 48.8 | 1590.6 +/- 37.7 | -33.7 +/- 1.1 |
| | | **GDRC** | 2841.4 +/- 9.9 | 1015.6 +/- 28.4 | 1724.9 +/- 31.4 | -21.0 +/- 0.5 |
| | $\omega = 0.4$ | PPO | 2763.7 +/- 29.8 | 1026.3 +/- 50.1 | 1064.7 +/- 29.1 | -31.7 +/- 1.2 |
| | | RE | 2599.8 +/- 50.5 | 1530.3 +/- 42.9 | 1548.3 +/- 20.7 | -44.9 +/- 1.1 |
| | | **GDRC** | 2642.7 +/- 51.5 | 1069.7 +/- 28.2 | 1462.6 +/- 36.6 | -25.7 +/- 1.0 |

# G ADDITIONAL RESULTS

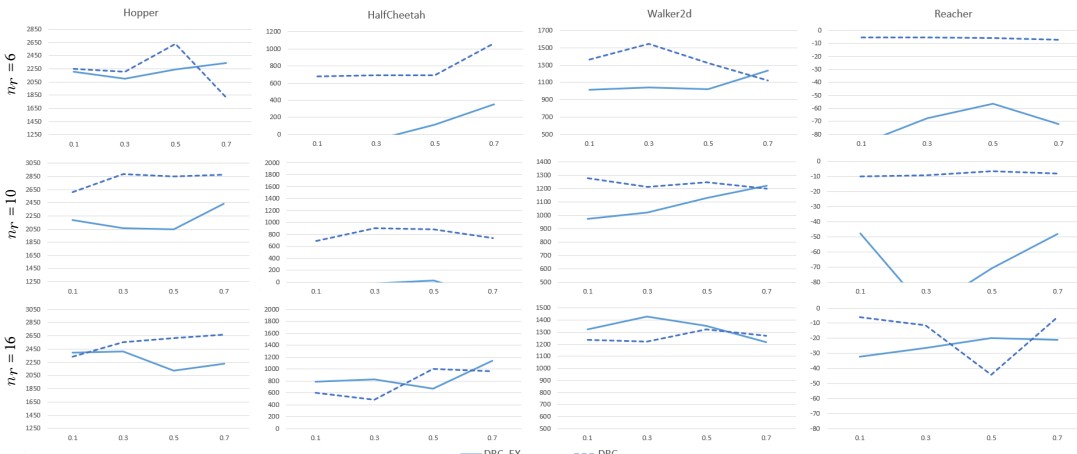

Figure 12: The results of Mujoco environments under GCM perturbations. DRC_EX represents using distributional reward critic by computing the expectation of the output. DRC is presented for comparison. The $x$-axis represents $\omega$. We keep the same scale of the y-axis as the previous charts for easy comparison.

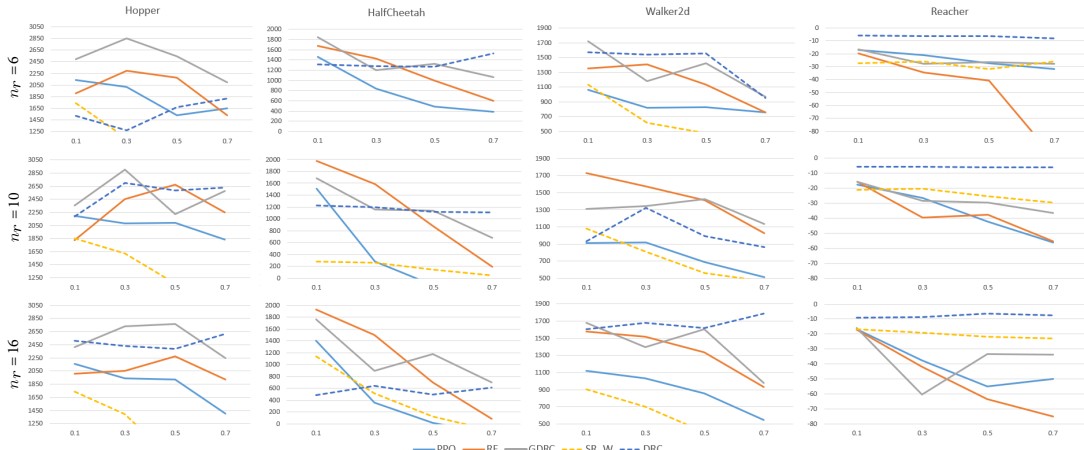

Figure 13: The results of Mujoco environments under GCM perturbations where the range of perturbed rewards gets doubled than the one before perturbations. Solid line methods can be applied without any information. **DRC** and **GDRC** are our methods. The $x$-axis represents $\omega$. We keep the same scale of the y-axis as the previous charts for easy comparison.

# H LEARNING CURVES

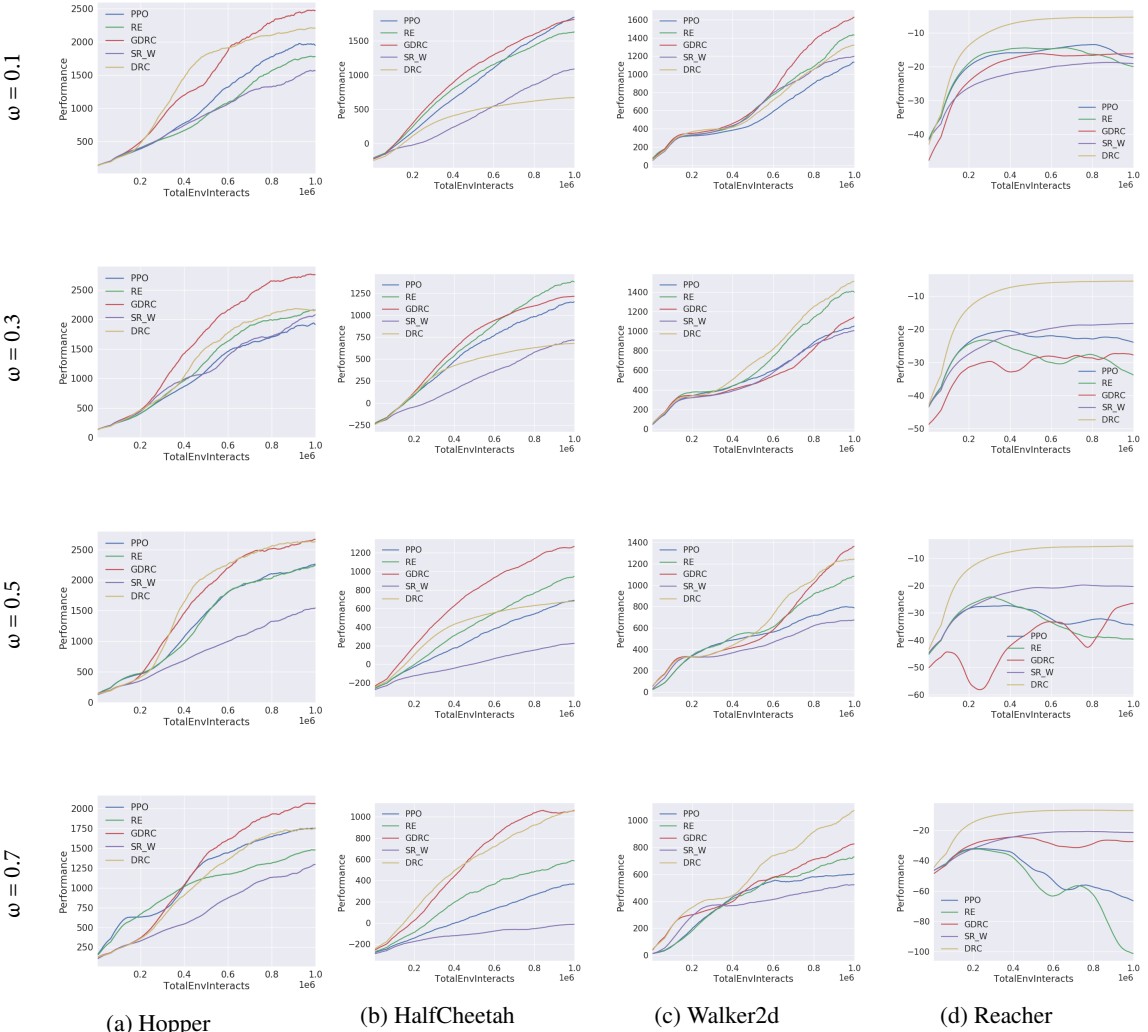

Figure 14: GCM Perturbations $n_r = 6$

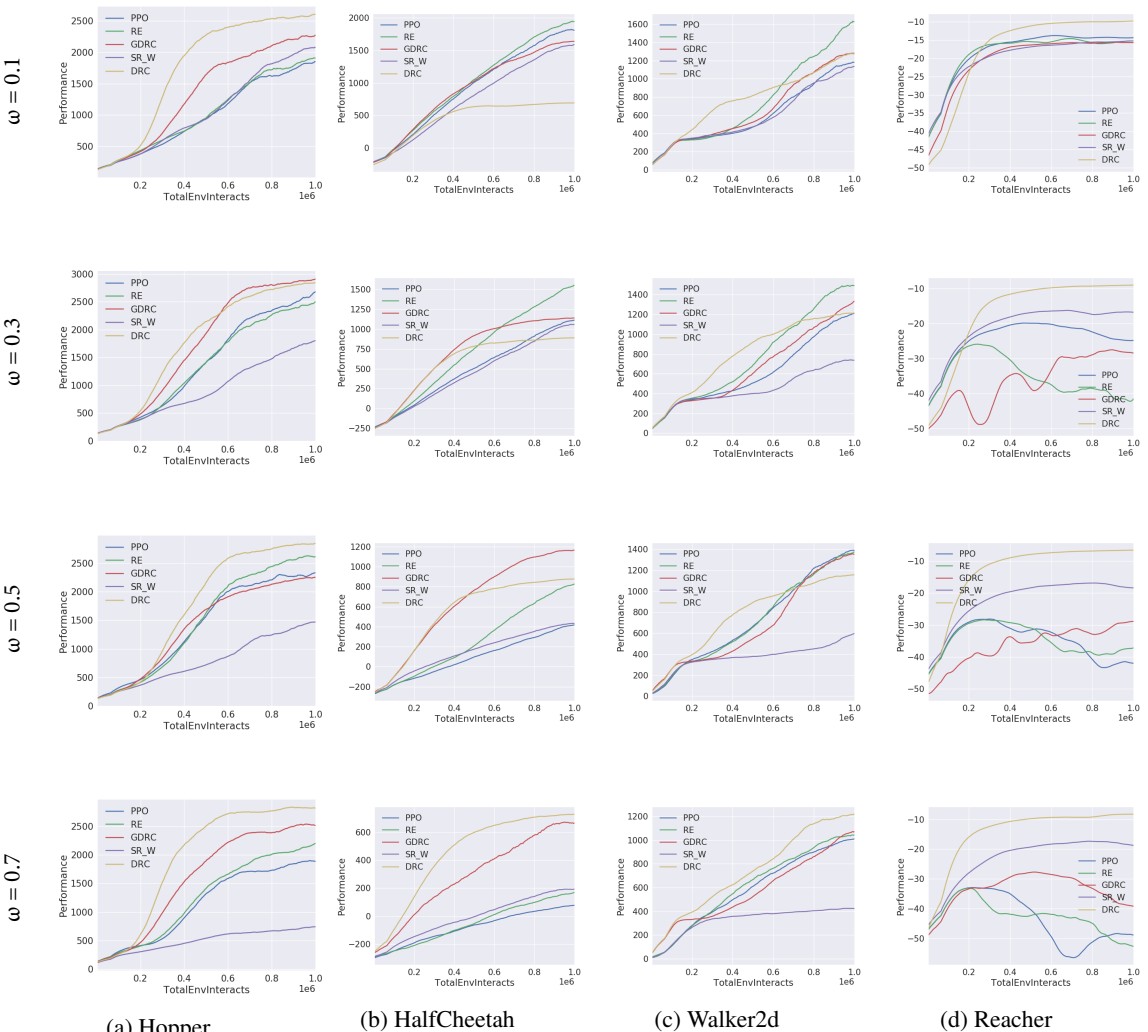

Figure 15: GCM Perturbations $n_r = 10$

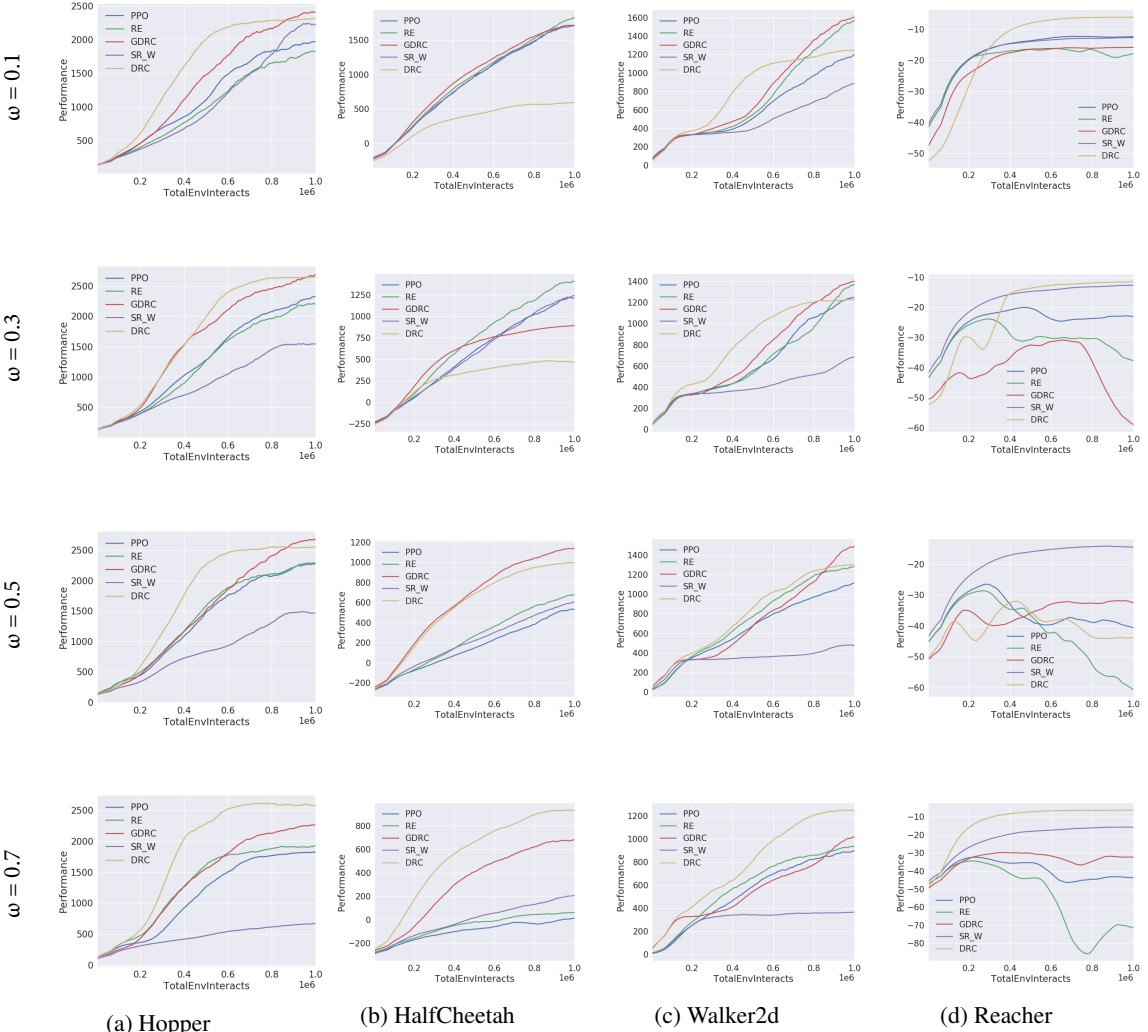

(a) Hopper  (b) HalfCheetah  (c) Walker2d  (d) Reacher

Figure 16: GCM Perturbations $n_r = 16$

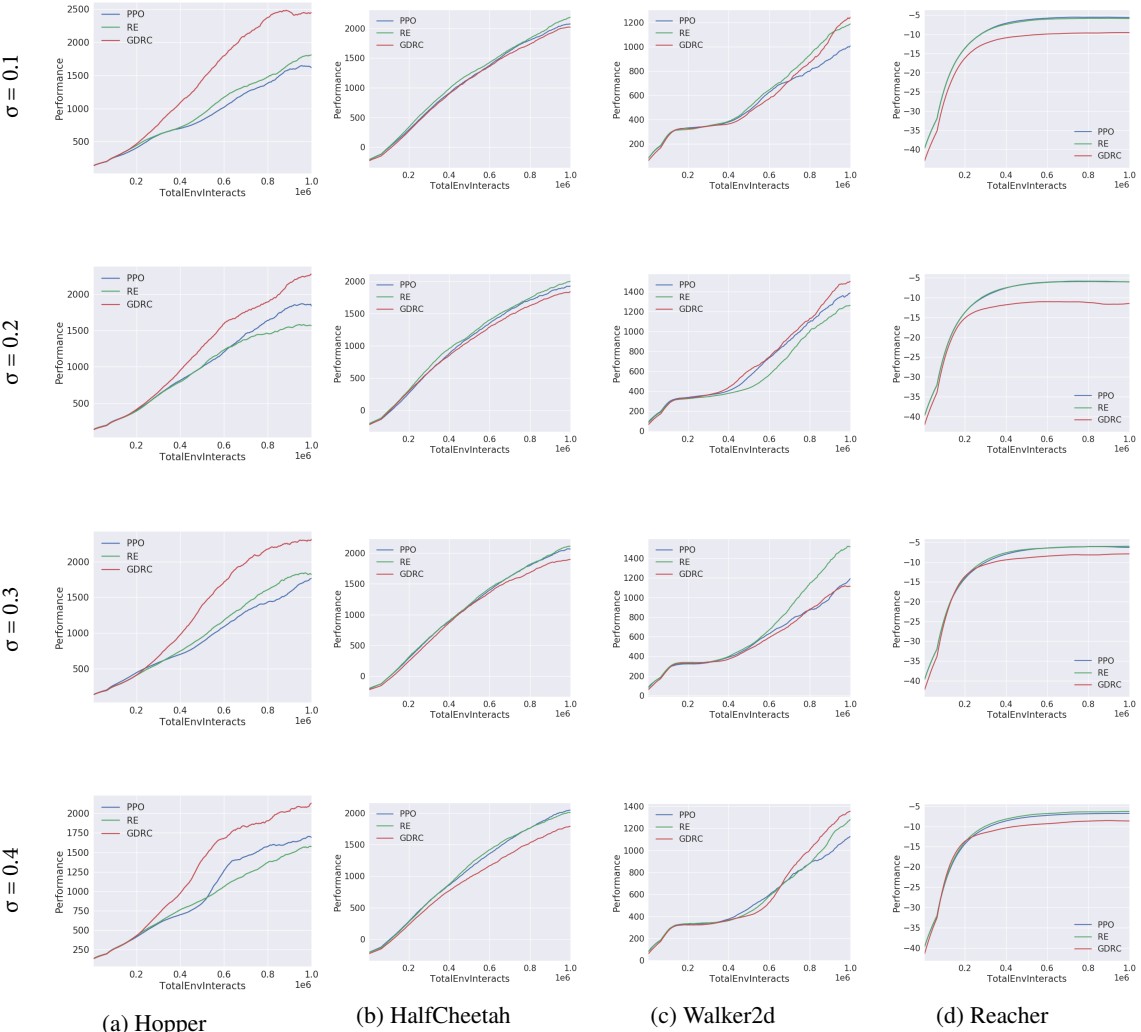

Figure 17: Continuous Perturbations (Gaussian Noise)

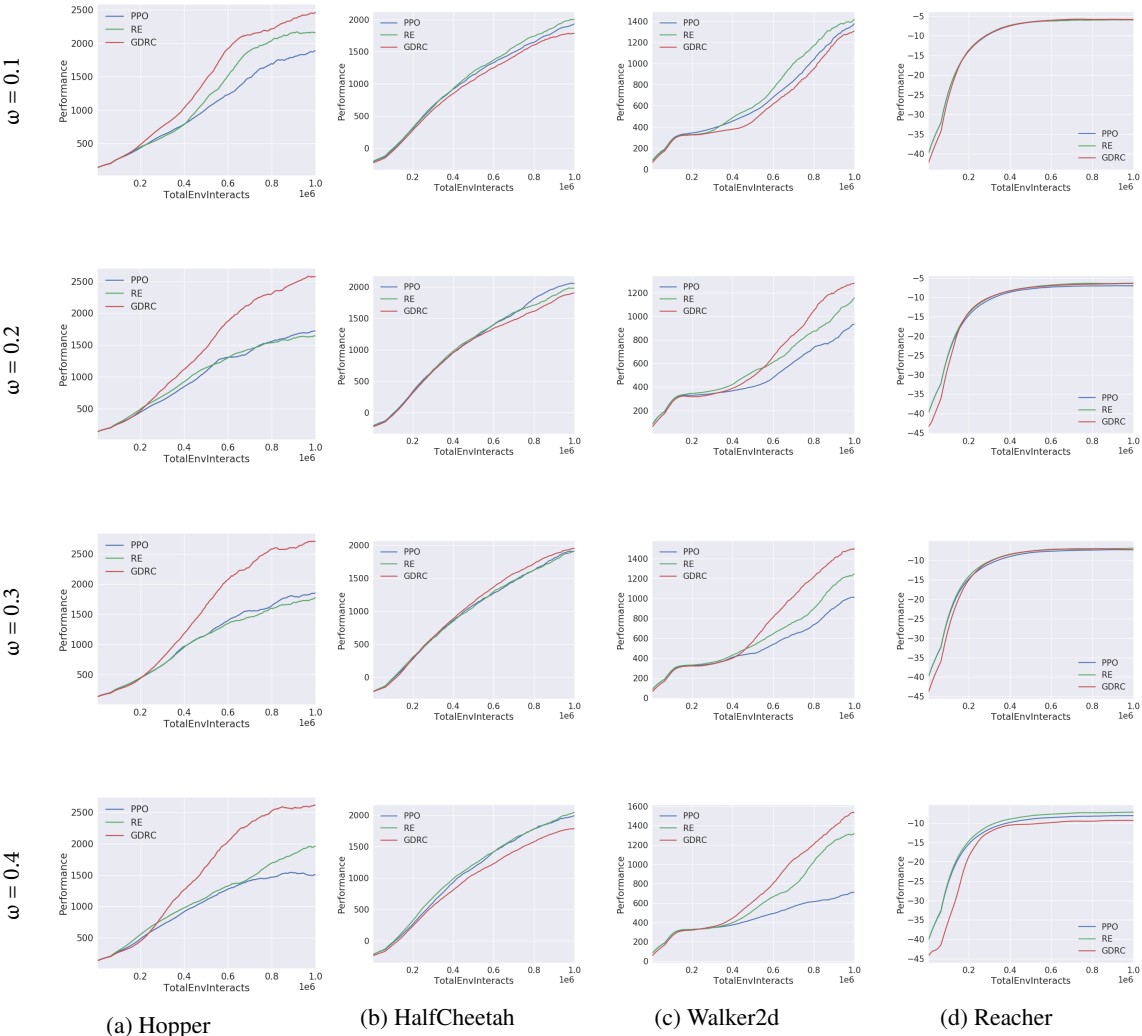

Figure 18: Continuous Perturbations (Uniform Noise)

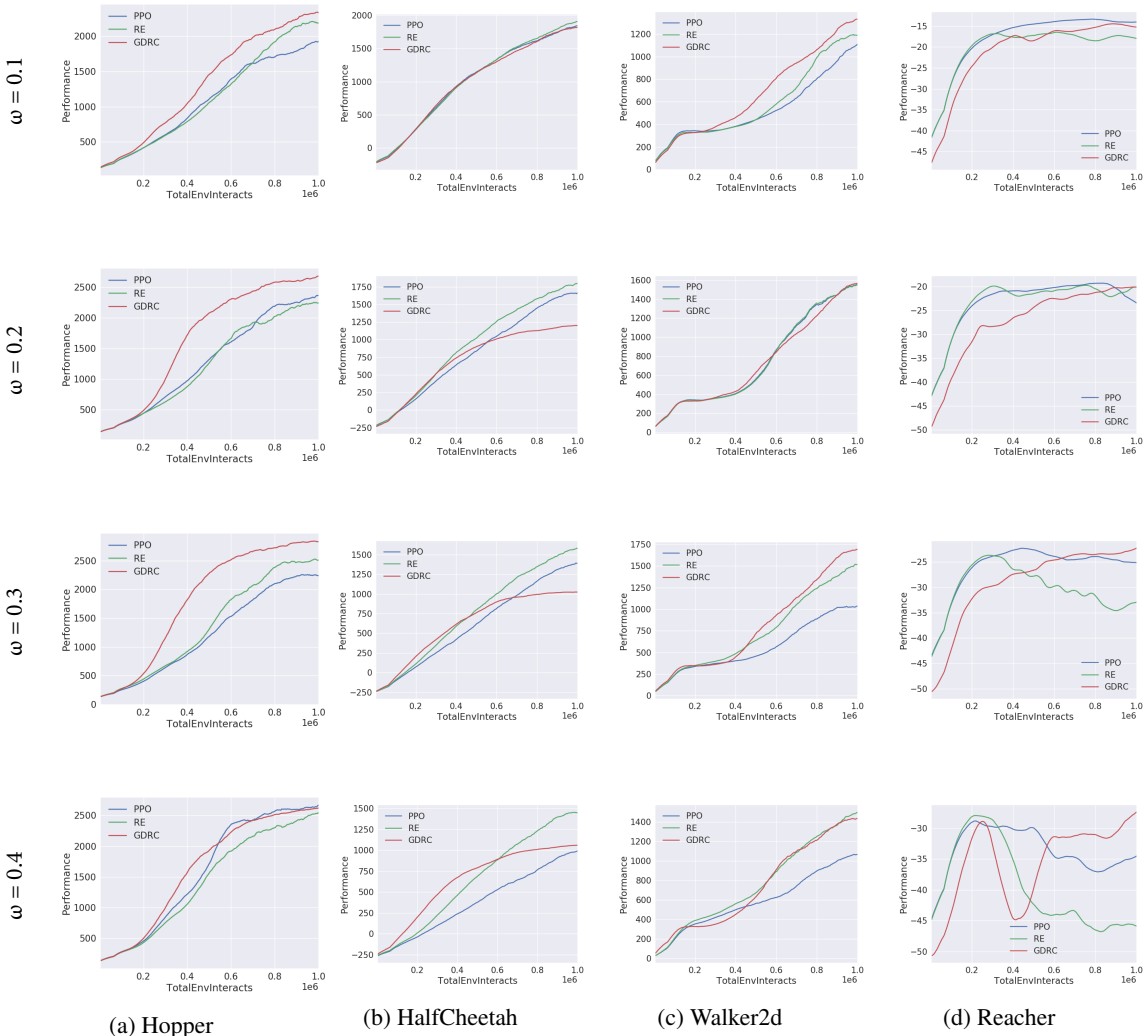

Figure 19: Continuous Perturbations (Reward Range Uniform Noise)

