# OpenReview forum: "The Distributional Reward Critic Architecture for Reinforcement Learning Under Confusion Matrix Reward Perturbations"
_ICLR.cc/2024/Conference — Submitted to ICLR 2024_

### Official Review · Reviewer_Jfuy · 2023-10-27

**Soundness:** 2 fair
**Presentation:** 2 fair
**Contribution:** 1 poor
**Rating:** 3
**Confidence:** 4

**Summary:**

The authors study RL algorithms under corrupted rewards. By considering the reward distribution and using the discretizing techniques, the confusion matrix can be better estimated by incorporating a reward critic. They also extend their methods to recover the noisy rewards from the known interval and partitions to the unknown one to cater to practical scenarios. Experiments are conducted on a few classical control and Mujoco environments.

**Strengths:**

* Considering the reward distribution in the corrupted reward setting is natural and well-motivated.

* The paper also studies the unknown interval and partition settings, which is more practical.

**Weaknesses:**

* **Potential concept misuse**. The reward is a naturally random variable and considering its distribution in the corrected reward setting is reasonable. However, it does not mean it is strongly correlated with distributional RL, which focuses on the return distribution instead of its expectation: value function.  I agree some density estimation techniques, such as categorical parameterization from C51 paper, can also be used here from distributional RL, but most of the technical issues, including the theoretical parts, between these two scenarios are very different. Emphasizing too much about the so-called distributional reward seems not natural.

* **Inaccurate theoretical statements**. Proposition 1 seems incorrect. Assume r equals $r_{min}$ and it is corrupted to be close to $r_{max}$, which exceeds the upper bound. Also, the proof in Appendix B.1 seems not complete, which I cannot directly follow. Also, Theorem 1 is a descriptive statement without rigorous mathematical statements. Thus, the results are not convincing in a rigorous way.

* **Limited contribution.** The confusion matrix is also well-studied in robust generative models, based on my knowledge, which can be naturally updated by a (critic) network. Thus, this contribution seems not sufficient. For avenues to cope with the continuous rewards, the discretization method this paper used is a direct application of C51 algorithm via the categorical parameterization. However, it seems the authors did not acknowledge that clearly. In distributional RL, C51 is typically inferior to quantile-based algorithms mainly due to its inability to handle unknown reward range and partitions. As far as I can tell, the paper mainly focuses on the known cases, with only heuristics strategies in the unknown settings. Hence, I can not recognize the sufficient contribution behind it and doubt its practical power.

* **Technical questions.** I am very confused by selecting the most probable reward label in step 7 of algorithm 1 instead of doing a reweighted sampling from the distribution. Note that if we only choose a single one, the optimal label tends to be deterministic rather than truly learning reward distributions. In the current version, along with the computation of the predicted reward value step, I think it is very similar to just doing the regression instead of a real multi-class classification since the $\arg\max$ does not consider the other probabilities when the reward belongs to other bins.
* **Empirical significance** Although the authors claim the empirical performance is desirable, I am afraid I cannot truly agree with that as I find most learning curves are not significantly better than other baselines instead of only making conclusions based on the final return. I may also have a question: although the authors claim that the proposed method can recover the reward better, is this benefit directly related to better performance? More importantly, Atari games are suggested to test the value-based RL algorithms.

* **Computation issues.** Learning a confusion matrix mapping is OK in generative models, but learning a reward critic (similarly viewed as a confusion matrix mapping) can be costly in RL as RL algorithms typically lack sample efficiency. Based on this knowledge, determining $n_o$ via the training loss is normally intractable, especially in a large environment with huge state and action spaces. Last but not least, using the training loss is a typical cross-validation strategy, which is straightforward and even trivial for me.

**Questions:**

Please refer to the weakness part.

### Minors.
* For continuous perturbations, the discretization technique should always induce a reconstruction error, which seems to have conflicts with some statements in the paper.
* The writing can also be improved as there are many grammar errors and typos.

---

> ### Author Response · Authors · 2023-11-17
> **(1/4) Rebuttal by Authors**
>
> We thank the reviewer for their review. Nevertheless, we believe they have misunderstood critical elements of the paper that are stated clearly.
>
> >Q1: Concept misuse of distributional RL
>
> A1: We do not think there is a concept misuse. As the reviewer says, distributional RL focuses on the return distribution. In our methods, our distributional reward critic focuses on reward distribution. "Reward Critic" is to predict the reward instead of value functions. "Distributional" means we apply the idea of distributional RL to predict the distribution instead of a single value as RE does. We would be happy to provide clarification if we have misunderstood the comment.
>
> >Q2: Proposition 1 seems incorrect. Assume r equals $r_{min}$ and it is corrupted to be close to $r_{max}$, which exceeds the upper bound.
>
> A2: Proposition is correct. Proposition 1 is to prove the coverage of the GCM setting by analyzing the maximum error of discretizing \bar_{r} into \tilde_{r}, where there is no discussion of the perturbation definitions. Please go to Section 3.2 for the definition of GCM. The field of the perturbed reward \tilde_{r} is $[r_{min}, r_{max}]$. We kindly request the reviewer to clarify what incorrect is with Proposition 1.
>
> >Q3: The proof in Appendix B.1 seems not complete.
>
> A3: The proof in Appendix B.1 is complete. The reviewer might not notice the first paragraph on Page 15.
>
> >Q4: Theorem 1 is a descriptive statement without rigorous mathematical statements.
>
> A4: The assertion that “the prediction from the distributional reward critic (DRC) is the distribution of the perturbed reward label” is a precise mathematical statement. This problem differs from conventional classification problems, where a given input (e.g., an image) has a fixed label. In our perturbed environment, a given state-action pair might have varying labels. It is difficult to prove the minimum of the loss function for all the samples directly so we wisely pick the samples of a state-action pair and prove the minimum of the loss function for these samples in its proof. Our goal is to prove what the output of the trained DRC using Cross-Entropy (CE) loss would be in our case.
>
> >Q5: The confusion matrix is also well-studied in robust generative models, which can be naturally updated by a (critic) network. Thus, this contribution seems not sufficient.
>
> A5: The general confusion matrix (GCM) perturbation is a whole concept. “Confusion Matrix” cannot be understood solely as its meaning has changed from its original one. The confusion matrix in GCM perturbations is to perturb the reward signals instead of being used to evaluate the performance. The confusion matrix perturbation is first used by Wang et al. to name their perturbation. We also have a detailed definition of GCM perturbations in Section 3.2.
> Besides, our distributional reward critic is not to update the confusion matrix but to predict the distribution of the perturbed reward labels of a given state-action pair. And the introduction of GCM perturbations is not our main contribution. Our main contribution is we propose a method (GDRC) that performs well in different settings without any information about the perturbation settings.
> We want the reviewer to clarify the relation between our work and the study of the confusion matrix in robust generative models. If we have misunderstood the comment, we would be happy to provide clarification if the review could elaborate further on it.

---

> > ### Author Response · Authors · 2023-11-17
> > **(2/4) Rebuttal by Authors**
> >
> > >Q6: The discretization method this paper used is a direct application of the C51 algorithm and the authors did not acknowledge that clearly. C51 is typically inferior to quantile-based algorithms.
> >
> > A6: We are confused by Q1 and Q6, where the reviewer first says we misuse the term distributional RL in Q1 and then says our work is a direct application of C51, one of the most popular works in distributional RL, in Q6. We have referenced the C51 paper in our Related Work section. We refuse to accept our work as a direct application of C51. According to the Related Work section, our work combines all the advantages of SR, RE, and C51 as below: SR uses a distributional approach in handling perturbations but does not incorporate neural networks, leading to underperformance and dependency on perturbation information; RE uses neural networks to predict rewards, but it is limited to order-preserving cases as its prediction is a single value; C51 predicts the value function distributionally, not the reward, and is not motivated by reward perturbation. Our work introduces a distributional reward critic (DRC) to predict rewards for handling reward perturbations, distinguishing it from a direct application of C51. More importantly, the number of bins of C51 is a hyper-parameter. Without knowing the reward discretization done by perturbations, there is no way for us to set it precisely if we use C51, which is what SR suffers under the Required Information group of Table 1 and what GDRC is to overcome.
> >
> > Regarding quantile regression, as we explain why we do not follow a quantile fitting form in the second paragraph of Page 2, we specifically choose fixed-width discretization to implement our method. In quantile fitting, we cannot control the width of each quantile as they are equal probability, leading to the uncontrollable reconstruction error while extracting the mode because we cannot control the width of each quantile. Besides, there is no way to decide the quantile number because we have no idea about the perturbation as we explain in the last paragraph, making the correction of the perturbed rewards even worse. In comparison, after realizing DRC using fixed-width discretization, we further resolve unknown reward range and partition issues in GDRC, which we consider as a significant contribution instead of a limitation. Therefore, we do not agree with our work being regarded as inferior just because we apply a C51-like discretization instead of quantile fitting.
> >
> > >Q7: The paper mainly focuses on the known cases, with only heuristics strategies in the unknown settings. Hence, I can not recognize the sufficient contribution behind it and doubt its practical power.
> >
> > A7: First, our primary objective at the very beginning is to address unknown settings, as evidenced by the Required Information group in Table 1. Second, the strategy of choosing $n_o$ in GDRC follows Theorem 2, proved in Appendix B.3, meaning our methods are not heuristic strategies.
> >
> > >Q8: I am very confused by selecting the most probable reward label instead of doing a reweighted sampling from the distribution. If we only choose a single one, the optimal label tends to be deterministic rather than truly learning reward distributions.
> >
> > A8: First, mode-preserving is the assumption of our methods. With mode-preserving, we can extract the most probable label as the predicted reward label. If we compute the expectation instead, it becomes a distributional version of RE, meaning we cannot recover the perturbations breaking the optimal policy. For the significance of the mode-preserving assumption, it is used by SR as well as shown in Table 1. Besides, mode-preserving is a weak assumption because our methods still work well under large noise ratios without a guarantee of mode-preserving under GCM perturbations and all the continuous perturbations people care about in RE necessitate mode-preserving shown in the Perturbations paragraph in Section 5.1. Moreover, GDRC even wins over RE under their targetted perturbations, showing mode-preserving is weak and GDRC is generally applicable. We clarify the motivation of mode-preserving in the last paragraph of Section 3 of the revised paper in blue.
> >
> > Second, we show the performance of DRC by computing the expectation instead of extracting the mode in Figure 12 in Appendix G for reference, from which we can tell DRC almost always performs much better than DRC_EX.
> >
> > Third, our method trains the distributional reward critic with observed rewards, learning the distribution of reward labels. When using rewards to train the critic and actor, we extract the reward label from the distribution using argmax and convert it into predicted reward values. Therefore, reward label distribution as the output from the distributional reward critic is learned.

---

> ### Author Response · Authors · 2023-11-17
> **(3/4) Rebuttal by Authors**
>
> >Q9: Most learning curves are not significantly better than other baselines instead of only making conclusions based on the final return.
>
> A9: First, we do not present any learning curves in the paper. Each point in Figures 5, 6, and 7 represents the average performance of 10 seeds running for 20 episodes each after training. We also provide learning curves in Figures 14-19 in Appendix H, where every line is the average of 10 seeds.
>
> Second, we do not agree that the outperformance is mostly insignificant. It is understandable the outperformance becomes more and more significant as we increase the noise ratio because any method handling perturbations inevitably suffers from performance loss. Besides, GDRC wins/ties the baseline methods in 40/57 (compared to 16/57 for the best baseline) of all kinds of perturbation settings (including small and large noise ratios) in Figures 5 and 6 as we present in the introduction, demonstrating the general outperformance of our methods. Even when considering the statistically significant win rate of GDRC (compared with PPO and RE) in Table 4, GDRC wins 29/48 cases with significance with the best baseline wins 10/48 cases with significance.
>
> Third, we unwrap the results more for the reviewer. In Figure 5, we win a small edge or achieve comparable performance as baseline methods when the noise ratio is small and the outperformance becomes larger as the noise ratio increases. In Figure 6, we can tell our methods almost always perform better than the baseline methods in Hopper and Reacher. As the noise ratio becomes larger, the outperformance becomes more significant as well. For Walker2d, the outperformance might not be that large although we almost always win over other baseline methods as well. This is because the benchmark performance is 1286.6 in Walker2d as we discuss in the second-last paragraph of Page 8, restricting the top performance we can achieve. What’s more, there are two factors rising from the original confusion matrix perturbations contributing to the not-too-bad performance of the baseline methods, which will be discussed in the next paragraph. For HalfCheetah, we win only when the noise ratio is large. We have a detailed analysis in the last paragraph of Page 8 and Figure 10, which is interesting and opens a door for further improvement. To summarize, we mostly win over the baseline methods and the outperformance gets more significant as the settings get perturbed more.
>
> Fourth, as we mention in the last paragraph, the two characteristics of the original confusion matrix perturbations lead to the relative not-too-bad performance of the baseline methods: the perturbed reward range remains unchanged as the one before perturbations, and the noise ratio is randomly assigned without intentional manipulations. The two factors together restrict the variance brought by the perturbation and cause less influence on the optimal policy. In reality, they can happen with high probability and the performance of the baseline methods will get destroyed severely, but our methods (DRC and GDRC) will not. To tell it in another way, our GCM perturbation settings can be more generalizable. We provide the results of the case the range of the perturbed rewards becomes twice as the clean ones, from $[r_{min}, r_{max}]$ to $[2r_{min}, 2r_{max}]$ where $r_{min}<0$ and $r_{max}>0$, in Figure 13 in Appendix G. GDRC wins/ties the best performance in 36/48 cases with larger outperformance. As for intentionally generating difficult GCM perturbations, we need to do statistics on the distribution of clean rewards, which we leave for future work.
>
> >Q10: Is this benefit directly related to better performance?
>
> A10: Yes, the only difference between our method and others lies in the use of different reward values for fitting the value function and improving the policy.

---

> > ### Author Response · Authors · 2023-11-17
> > **(4/4) Rebuttal by Authors**
> >
> > >Q11: Atari games are suggested to test the value-based RL algorithms
> >
> > A11: First, we have evaluated our methods using DQN, one of the value-based RL algorithms, in Figure 5.
> >
> > Second, as we state in the Introduction, our methods (DRC and GDRC) are effective in all kinds of settings (e.g. different algorithms, environments, perturbations, etc.) as long as mode-preserving holds.
> >
> > Third, I want to clarify our ideas on designing the experiments. We want to evaluate the methods from many perspectives thoroughly to demonstrate the effectiveness of our proposed methods.
> > * We include control tasks as Wang et al. did to evaluate SR as it is not applicable to complex environments regarding state-action space as SR needs to discretize state-action space to recover the confusion matrix.
> > * We include both off-policy algorithms (DDPG and DQN) and on-policy algorithms (PPO), both value-based algorithms (DQN) and actor-critic algorithms (DDPG and PPO).
> > * We include Mujoco environments because of two reasons. First, the four Mujoco environments are what RE experiments with. Second, moving from a small state-action space (e.g. control tasks), where the confusion matrix could be recovered, to a large one (e,g. Mujoco) is a big challenge for SR, for us to overcome as well.
> > * We include both two kinds of perturbations (general confusion matrix perturbations in Wang et al. and continuous perturbations in Romoff et al.)
> > * We include another dimension, $n_r$, of GCM perturbations for evaluation, which has not been evaluated by Wang et al.. Only by doing this, we can analyze the conditions the methods implicitly need to work in Section 5.2 so that we have some interesting observations in the last paragraph and then propose some potential solutions for further improvement in the Future Work section.
> >
> > To summarize, we do try our best to evaluate all the methods as thoroughly as possible and there is no time for us to set up the experiments in another setting.
> >
> > >Q11: Learning a reward critic can be costly in RL as RL algorithms typically lack sample efficiency
> >
> > A11: Actually, lacking sample efficiency in RL contributes to the correct prediction from the distributional reward critic. Our methods for training the distributional reward critic use the same number of samples of standard RL training processes. According to Figure 8, the numbers of the samples used to train the distributional reward critic in P2 and used to train the critic and/or actor are the same. Indeed, the lack of sample efficiency in training an RL agent is really helpful for the training of our distributional reward critic. This is because many samples collected for training the agent are not that helpful in converging to the optimal policy quickly, but all the collected samples almost have the same value for training the distributional reward critic. To tell it from another perspective, the agent might be stuck in a local area of the state-action space for a long time, which is not good for policy improvement but good for the correct prediction of the distributional reward critic.
> >
> > >Q12: Determining n_o via the training loss is normally intractable, especially in a large environment with huge state and action spaces. Using the training loss is a typical cross-validation strategy, which is straightforward and even trivial for me.
> >
> > A12: As mentioned in A7, we believe the feasibility of our methods has been sufficiently demonstrated. One thing to notice is that we do use the CE, but we use it wisely and differently from what people always do. Using the CE loss to choose an appropriate $n_o$ has been proved through four steps as discussed in A7. To address the limitation of recovering the confusion matrix for SR in large environment spaces, we incorporate the distributional reward critic, a neural network, to resolve this issue. There is nothing to do with cross-validation regarding how we use the CE. We want the reviewer to clarify the relation between using the CE loss for cross-validation and our work.
> >
> > >Q13: For continuous perturbations, the discretization technique should always induce a reconstruction error, which seems to conflict with some statements in the paper.
> >
> > A13: All of the analysis of our methods analysis occurs under the GCM perturbations as clearly stated in the method section. Continuous perturbations are not our targeted ones in designing the methods, where we want to demonstrate the broad applicability of our approaches and the weakness of the mode-preserving assumption by showing their general application even under untargeted perturbations.
> >
> > >Q14: The writing can also be improved as there are many grammar errors and typos.
> >
> > A14: We will revise the paper carefully as suggested. We welcome feedback from the reviewer on areas requiring corrections, allowing us to make targeted improvements.

---

> > > ### Author Response · Authors · 2023-11-20
> > >
> > > Again, we thank the reviewer for their valuable review and hope their concerns are addressed properly. If they are satisfied with the rebuttal, we kindly request the reviewer to raise their score. If not, we would be happy to continue the discussion during the rebuttal.

---

> > > > ### Comment · Reviewer_Jfuy · 2023-12-01
> > > >
> > > > I acknowledge that I have read the rebuttal. Some issues have been clarified, but I still do not fully agree with many critical points from the authors' responses. Therefore, I keep my score for now.

---

### Official Review · Reviewer_wJ8T · 2023-10-30

**Soundness:** 3 good
**Presentation:** 3 good
**Contribution:** 2 fair
**Rating:** 6
**Confidence:** 3

**Summary:**

This paper studies reinforcement learning with perturbed reward, where the reward perceived by the agent is injected with noise. Inspired by previous work, this paper proposes a new form of reward perturbation based on a confusion matrix, which shuffles the rewards of discretized intervals and can preserve local continuity within each interval. Under the mode-preserving assumption, which guarantees the true reward will be observed the most frequently, the paper proposes to learn a classifier to identify the interval index of the true reward when the discretization is known. When the number of intervals or even the reward range is unknown, the paper proposes to use an ensemble or a streaming technique to address the respective unknown parameter. Experimental results on four continuous control tasks show the effectiveness of the proposed techniques.

**Strengths:**

The strength of the paper is that it provides a comprehensive study of the proposed problem. As stated in the above summary, to address the new form of confusion matrix perturbation, the paper proposes a new algorithm as well as extends the new algorithm to handle scenarios in which some/all parts of the discretization are unknown. In addition, it also provides the guarantee that the correct reward can be predicted asymptotically and the theoretical justification for the voting mechanism of the ensemble approach when the number of intervals is unknown.

In terms of presentation, the paper is well-written and organized.

**Weaknesses:**

In my opinion, the major potential weakness of the paper is that it has a restrictive type of perturbation. The paper assumes the rewards in different intervals are shuffled and the true reward is still the most frequently observed (mode-preserving property). However, I find this assumption too artificial and not natural. On the other hand, while the paper claims the assumption of RE (Romoff et al. 2018) to be stringent, I found it to be more natural. I understand the argument that RE can’t properly handle non-order-preserving perturbation. Can you provide more justification for this assumption? Also, how robust would the method be if GCM is not mode-preserving?

**Questions:**

See Weaknesses for major questions.

Clarification questions or typos that do not impact assessment:
What is SR_W? How is it different from SR?
The reference style is a bit messy and inconsistent throughout the paper.
On page 3, by convention, $\beta : \Delta(S)$ should be $\beta \in \Delta(S)$.

---

> ### Author Response · Authors · 2023-11-17
> **Rebuttal by Authors**
>
> We thank the reviewer for their valuable review. Here are our answers to the questions:
>
> >Q1: The mode-preserving assumption is too artificial and not natural. The assumption of RE is more natural.
>
> A2: Mode-preserving is the assumption of the surrogate reward (SR) method instead of being created by us as shown in Table 1. When recovering the confusion matrix, they use majority voting to decide the true reward of the samples for a state-action pair assuming mode-preserving. Besides, the forms of $\tilde{R}$ of the continuous perturbation functions (e.g. Gaussian perturbations) people pay attention to in RE necessitate mode-preserving although the space is continuous, which shows mode-preserving is a weaker assumption in the cases people are interested in. Following this idea, we propose a method that combines the advantages of both SR and RE under the assumption of mode-preserving.
>
> One thing to notice is that we do not guarantee mode-preserving in the generation of GCM we experiment with. Nevertheless, our methods perform effectively even under high noise ratios (e.g., $\omega =0.7$), demonstrating that our assumption of mode-preserving is weak and our methods are robust. Furthermore, our method outperforms baseline methods even under the perturbation (continuous perturbation) tailored for RE shown in Section 5.4, indicating our method's broad applicability and mode-preserving is a weak assumption again.
>
> In a word, with a weak mode-preserving assumption, our methods are more broadly applicable under many kinds of perturbations. We clarify the motivation of mode-preserving in the last paragraph of Section 3 of the revised paper in blue.
>
> >Q2: How robust would the method be if GCM is not mode-preserving?
>
> A2: Any method dealing with perturbations needs an assumption to work. If mode-preserving, the only assumption of our methods, does not hold, there is no way for a method to work theoretically because we have no assumption about the perturbation for our reference to recover the clean rewards.
>
> >Q3: Clarification questions: How is SR_W different from SR? The reference style is a bit messy and inconsistent throughout the paper. On page 3, by convention, $\beta:\Delta(S)$ should be $\beta\in\Delta(S)$.
>
> A3: SR can learn the confusion matrix within a simple environment space (e.g. Pendulum), but SR_W needs to know the whole confusion matrix before being applied, which is the last sentence of the first paragraph of Page 3. We uniform the use of the reference style and modify the notation as suggested.

---

> > ### Author Response · Authors · 2023-11-20
> >
> > Again, we thank the reviewer for their valuable review and hope their concerns are addressed properly. If they are satisfied with the rebuttal, we kindly request the reviewer to raise their score. If not, we would be happy to continue the discussion during the rebuttal.

---

> > ### Comment · Reviewer_wJ8T · 2023-11-22
> >
> > Thank you for your answers and clarifications. Your answers to the questions have helped me understand your argument better. While Gaussian noise is mode-preserving, the order-preserving perturbation, in general, is not necessarily mode-preserving (for example, when there is a non-zero shift in the reward). Thus, I don’t perceive mode-preserving as a weak assumption compared to the order-preserving assumption. They are not directly comparable. However, I do agree that mode-preserving is a useful assumption.
> >
> > As for the example with a high noise ratio ($\omega > 0.7$), the true reward will still be the most presented with a high probability. In other words, the example will likely be mode-preserving. Thus, I don’t consider this as support for the argument that the proposed methods are robust and work even when the assumption does not hold. Including results with total perturbed environments ($\omega = 1$) may help to provide the support.
> >
> > After reading other reviews, I noticed the concern Reviewer XsR2 raised that the method only works for MDPs with deterministic rewards, while it may suffer from unavoidable bias when the rewards are stochastic. I think the paper should be upfront about the potential bias introduced by ignoring the stochasticity of the rewards. I think incorporating the adaptation of the proposed methods to the stochastic rewards case would make the paper much stronger.
> >
> > Overall, I retain my positive assessment of the paper for now.

---

> ### Author Response · Authors · 2023-11-22
>
> We thank the valuable reply from the reviewer. We just want to add some potentially useful information for the reviewer’s further reference.
>
>
> >Point 1: While Gaussian noise is mode-preserving, the order-preserving perturbation, in general, is not necessarily mode-preserving (for example, when there is a non-zero shift in the reward).
>
> Reply 1: That is a very good point. Indeed, our assumption can be even weaker as the mode of the perturbed rewards follows a positive affine transformation with some conditions. In comparison, RE follows that the expectation of the perturbed rewards follows the same rules. RE also cannot correct the systematic bias, but it works whenever the bias does not influence the optimal policy. To make the story simpler, we do not talk too many details.
>
>
> >Point 2: Thus, I don’t perceive mode-preserving as a weak assumption compared to the order-preserving assumption. They are not directly comparable. However, I do agree that mode-preserving is a useful assumption.
>
> Reply 2: We thank the reviewer for recognizing the usefulness of the mode-preserving assumption. Actually, all the baseline methods do not explicitly state their applicable scenarios and needed assumptions. Instead, we analyze them thoroughly and present the assumption of our methods clearly. We agree they might not be directly comparable, but the outperformance of our methods under two kinds of perturbations might support the broader application of our methods.
>
>
> >Point 3: The example ($\omega=0.7$) will likely be mode-preserving. Thus, I don’t consider this as support for the argument that the proposed methods are robust and work even when the assumption does not hold. Including results with total perturbed environments ($\omega=1$) may help to provide the support.
>
> Reply 3: We do not propose our methods work even when the assumption does not hold. We want to convey that our methods even work under large perturbations even without guaranteeing mode-preserving, showing mode-preserving is not a stringent assumption and our methods are robust. As we answered in A2 above, our methods do not work when mode-preserving does not hold, which is the same case for the other methods if their assumptions do not hold.
>
>
> >Point 4: The method only works for MDPs with deterministic rewards, while it may suffer from unavoidable bias when the rewards are stochastic. I think the paper should be upfront about the potential bias introduced by ignoring the stochasticity of the rewards. I think incorporating the adaptation of the proposed methods to the stochastic rewards case would make the paper much stronger.
>
> Reply 4: Incorporating the adaptation of the proposed methods to the stochastic rewards case is a very good point. We want to clarify the motivation of the methods of handling perturbations. In MDPs with deterministic rewards, which is always the case for SR, RE, and ours, the methods of handling perturbations aim to correct the stochastics of the perturbed rewards because of perturbations (e.g., human errors, sensor noise, etc.). The MDPs with random rewards are not the targeted MDPs of our three works, meaning the stochastics inside the random rewards are not what we want to correct. As for the performance of different methods, it depends on their assumptions and the method designs. In the defined MDPs, which are also what people usually define, of the three methods, ours is the best not only from the theoretical perspective but from the experimental one.
>
> However, we agree to discuss the adaptation of our methods in another kind of MDP. It might be hard to combine them together directly in this work as they are under two kinds of MDPs, but we promise to have a separate part to discuss the adaptation of our methods in MDPs with random rewards.
>
>
> Again, we really appreciate the insightful thoughts from the reviewer.

---

### Official Review · Reviewer_XsR2 · 2023-10-31

**Soundness:** 2 fair
**Presentation:** 3 good
**Contribution:** 2 fair
**Rating:** 5
**Confidence:** 4

**Summary:**

The authors consider the problem of training a reinforcement learning agent under a perturbed reward signal. Similar to Romoff et al. (2018), they propose to learn a reward estimator and then train the agent from the predicted rewards rather than the environment rewards. However, unlike Romoff et al., who strive to learn the expected reward, the authors propose to learn the reward distribution (assuming a categorical distribution), and then train the RL agent from the mode of the predicted distribution. This purportedly allows the approach to handle a broader range of reward perturbations, including those that are not order-preserving.

**Strengths:**

The idea of predicting the reward via a categorical distribution is novel, and it's an interesting adaptation of ideas from distributional RL. One thing I like about the approach is that, unlike the C51 agent (Bellemare et al., 2017), the number of intervals under GDRC is adaptive. The analysis in Section 5.2 supporting this adaptive approach is very nice. In fact, on the whole, all of the extra analysis that is done beyond the headline results is quite helpful. The authors are upfront about some of the shortcomings of the approach, e.g., the mode collapse problem described at the end of page 8 (where they again include extra, helpful analysis in the Appendix). While the content of the paper is fairly technical, it's mostly well-presented and I think I understood most of the main details. Lastly, I agree that this is an important research area, especially given that finetuning LLMs via RL with human feedback has become such a hot topic recently.

**Weaknesses:**

One desirable property of any approach to this problem is that it should never decrease performance on tasks where the rewards are *unperturbed*. However, this doesn't hold for the proposed approach. For example, consider a simple betting game where the reward of some bet is -1 with probability 0.9 and +10 with probability 0.1. Romoff et al.'s approach will (in theory) learn that the expected reward is 0.1, and thus the agent will learn that the bet is worth taking. On the other hand, the proposed approach will learn that the mode reward is -1, and hence the agent will learn to avoid the bet. Generalising from this, if we want the approach to be "safe" then it ought to preserve the expected reward.

If we somehow knew that the clean rewards were deterministic and the perturbation was mode-preserving then this would no longer be an issue, but this is a big assumption and it's hard to see how we'd ever know this in a realistic application. For example, human feedback for LLMs arguably is stochastic, not just "perturbed", since different humans have different preferences.

Less pressing issues:

- The difference between the proposed approach and Romoff et al.'s can be broken into two parts: (1) Learning a categorical estimate, rather than a scalar one; (2) The agent is trained from the mode reward, rather than the mean. I'd like to see a comparison versus an approach that uses (1) but not (2), i.e., the reward distribution is learned, but the agent is trained from the mean of the distribution. This would disengangle how much of the performance impact comes from assuming a categorical distribution and how much comes from assuming that the perturbation is mode-preserving.
- Page 8 states: "The agents in Hopper and Walker2d are encouraged to live longer because of the positive expectation of perturbed rewards". However, this contradicts the statement on page 2 that affine transformations preserve optimal policies. The statement on page 2 should be refined to say that this only holds for non-episodic tasks.
- In my opinion, the proof of Theorem 1 is unnecessary and just overcomplicates the paper. All it really says is "under perfect conditions, the cross entropy loss will go to zero, and hence the approach learns what it's supposed to". We'd have AGI already if he had "sufficiently expressive" neutral networks + unlimited training samples + the necessary compute :p

**Questions:**

I was slightly confused by the statement on page 3: "we assume that the distribution of the perturbed reward depends only on the true reward, i.e., $\tilde{r}_t \sim \tilde{R}(R(s_t, a_t))$". Couldn't the proposed approach handle state- and action-dependent perturbations too?

---

> ### Author Response · Authors · 2023-11-17
> **(1/2) Rebuttal by Authors**
>
> We thank the reviewer for their valuable review. Here are our answers to the questions:
>
> >Q1: One desirable property of any approach to this problem is that it should not decrease performance on tasks where the rewards are unperturbed. Romoff et al.'s approach works for the betting game, but the proposed approach does not.
>
> A1: It is overly stringent to expect a method designed for dealing with perturbations to not suffer from any performance loss in a clean environment. If there is always no degradation in performance for handling perturbations, there is no need for the original method.
>
> In the definition of MDP in the first paragraph of Section 3, the reward $r_t$ is from states and actions $(s_t, a_t)$ expressed as $r_t = R(s_t, a_t)$, but the reward $r_t$ is also allowed to depend on the transition, meaning $r_t = R(s_t, a_t, s_t^\prime)$. To make the betting game MDP, we need to include $s_t^\prime$ as the input to the reward function to tell whether we win or lose if we choose to play the betting machine. Otherwise, it even does not accord with the definition of MDP as the same state-action pairs can generate different rewards. To handle the betting game, there is only a small adaption needed – input $s_t^\prime$ to the distributional reward critic other than $(s_t, a_t)$. Therefore, our methods also work in the betting game.
>
>
> >Q2: If we knew that the clean rewards were deterministic and the perturbation was mode-preserving, it would no longer be an issue, but mode-preserving is a significant assumption, and it's hard to see how we'd ever know this in a realistic application. For example, human feedback for LLMs is arguably stochastic and not just "perturbed," as different humans have different preferences.
>
> A2: First, deterministic clean rewards are not needed for our method. In the environments we experiment with, without considering any randomness in the environments, the rewards are indeed fixed given state-action. However, the clean rewards are not static because of the randomness.
>
> Second, there is still a big issue even if mode-preserving holds. Perturbations can introduce variance, slowing the process of value function fitting, which is what RE addresses. GCM can induce bias breaking the preferences of the taken actions (order-breaking bias), directly affecting policy performance by altering the optimal policy measured by the perturbed rewards.
>
> Third, any work dealing with perturbation invariably requires assumptions of perturbations, whether mode-preserving or order-preserving. Mode-preserving is not created by us, which is the assumption of SR as shown in Table 1. When recovering the confusion matrix, they use majority voting to decide the true reward of the samples for a state-action chunk while assuming mode-preserving. Besides, the forms of $\tilde{R}$ of the continuous perturbation functions (e.g. Gaussian perturbations) people pay attention to in RE necessitate mode-preserving although the space is continuous, which shows mode-preserving is a weaker assumption in the cases people are interested in. Also as we discuss in the second paragraph of A2, our methods can deal with non-order-preserving perturbations with mode-preserving assumption but RE cannot with a positive affine transformation. One thing to notice is that we do not guarantee mode-preserving in the generation of GCM we experiment with. Nevertheless, our methods perform effectively even under high noise ratios (e.g., $\omega =0.7$), demonstrating that our assumption of mode-preserving is weak and our methods are robust. Furthermore, our method outperforms baseline methods even under the perturbation (continuous perturbation) tailored for RE shown in Section 5.4, indicating our method's broad applicability and mode-preserving is a weak assumption again.  We clarify the importance of mode-preserving in the last paragraph of Section 3 of the revised paper in blue.
>
> Fourth, regarding RLHF, we still need to have some pre-knowledge of the distribution of perturbed labels, and then apply appropriate methods to handle perturbations. In RLHF, the samples collected from humans are usually with discrete labels (e.g. integers from 1 to 5), which can be regarded as problems dealing with noisy labels. Mode-preserving is the most used assumption for the methods to handle noisy labels except for the case you have other assumptions of the label distributions. Even if the assumption/pre-knowledge is about the distribution instead of the mode, our methods can still be adapted correspondingly as they can predict the perturbed labels, but RE cannot because the information on the distribution is not kept. Within the discrete space of RLHF, the outperformance of our methods is more significant compared with the baseline methods.
>
> In a word, we want to clarify an assumption is needed for any work dealing with perturbation and mode-preserving is a weak one, which explains our methods work under many kinds of perturbations.

---

> > ### Author Response · Authors · 2023-11-17
> > **(2/2) Rebuttal by Authors**
> >
> > >Q3: Disentangle the factor of using a categorical distribution contributing to the performance improvement.
> >
> > A3: This is a good question. If we calculate the expectation, it essentially becomes a distributional version of RE, making it impossible to recover the perturbation where the assumption of a positive affine transformation does not hold. We include the experiment results in Figure 12 in Appendix G for verification, from which we can tell DRC almost always performs much better than DRC_EX. Therefore, we conclude extracting the mode is the most important factor contributing to the performance improvement, instead of a categorical distribution fitting.
> >
> > >Q4: Page 8 states: "The agents in Hopper and Walker2d are encouraged to live longer because of the positive expectation of perturbed rewards". However, this contradicts the statement on page 2 that affine transformations preserve optimal policies. The statement on page 2 should be refined to say that this only holds for non-episodic tasks.
> >
> > A4: There is no contradiction. The two statements are under different perturbations. Under GCM perturbations, the statement on Page 8 shows the agent can still get some information from the expectation of fully perturbed reward signals because the perturbed reward range stays unchanged after the perturbation. It is to explain the not-too-bad performance of the baseline methods under GCM perturbations. Under continuous perturbations, the statement on Page 2 is about the assumption of RE expressed as $\mathbb{E}(\tilde{r})=\omega_0\cdot r +\omega_1(\omega_0 > 0)$, which ensures the unchanged optimal policy for both episodic and non-episodic tasks after the perturbation.
> >
> > >Q5: The proof of Theorem 1 is unnecessary.
> >
> > A5: The reviewer’s understanding is partially incorrect. In typical classification problems, an input (e.g., an image) has a definite label. However, samples of the same state-action pair can have varying labels in our perturbed environment. It is difficult to prove the minimum of the loss function for all the samples directly so we wisely pick the samples of a state-action pair and prove the minimum of the loss function for these samples. Our goal is to demonstrate that the output of the trained distributional reward critic corresponds to the distribution of perturbed reward labels by using Cross-Entropy (CE) loss. The reviewer misunderstand the CE will go to zero as the CE does not converge to zero because of perturbation, evident from the proof of Theorem 1, Figures 2, and Figure 4, where the finally converged output is the distribution of perturbed reward labels. It might cause such misunderstandings if we do not emphasize it enough.
> >
> > >Q6: Could the proposed approach handle state- and action-dependent perturbations as well?
> >
> > A6: Yes, our method can address state- and action-dependent perturbations, provided that the mode-preserving assumption holds. The reward-dependent (state-action dependent) perturbation is just the perturbation we experiment with for evaluation. How the perturbation is constructed does not impact our method at all. We experiment with this kind of perturbation because it is commonly used and the general confusion matrix (GCM) perturbation is general.

---

> > > ### Comment · Reviewer_XsR2 · 2023-11-17
> > >
> > > Thanks for your response, I appreciate the inclusion of the DRC_EX results, which look convincing.
> > >
> > > Regarding Q5, you're right, I should have said that the loss will be "minimised", not "go to zero". My point really was that the proof amounts to "under unrealistically perfect conditions, the approach works", which isn't surprising IMO and just over-mathsifies the paper. I'm not staunchly opposed to including it though; I guess it could be helpful for some readers.
> > >
> > > > Under continuous perturbations, the statement on Page 2 is about the assumption of RE expressed as $\mathbb{E}(\tilde{r}) = \omega_0 \cdot r + \omega_1 (\omega_0 > 0)$, which ensures the unchanged optimal policy for both episodic and non-episodic tasks after the perturbation.
> > >
> > > This isn't true for episodic tasks. Consider an environment where an agent can choose to continue or terminate the task. If all rewards are positive, the optimal policy is to continue, but if they're shifted to negative values by setting $\omega_1 \lt 0$ then it's optimal to terminate the task. This isn't a major issue, but I'd suggest correcting the sentence.

---

> > > > ### Author Response · Authors · 2023-11-17
> > > >
> > > > We thank the reviewer for their quick reply. Here are the answers to their questions:
> > > >
> > > > >Q1: I thought that the main assumption in the problem setting was that the same state-action pairs could generate different rewards. Is the assumption that the clean rewards are deterministic but the perturbed rewards aren't? If so, what's the justification for this?
> > > >
> > > > A1: Yes, the same state-action pairs could generate different rewards under perturbations. “The definition of MDP” corresponds to the ones used in SR and RE papers and we follow that definition, whose reward function is $R(s, a)$ instead of a random variable. Yes, the clean rewards of the environments we (SR, RE, and us) consider are deterministic without any stochastics (e.g. sensor noise), which is the property of those environments, but the rewards can be also stochastic with stochastics in the environment.
> > > >
> > > > We want to clarify the difference between our environments and the betting game, which is like a bandit problem. First, without any stochastics, the betting game environment can generate random rewards, meaning the reward is a random variable. Second, even with stochastics, the rewards of the state-action change continuously instead of the very different rewards of the betting game.
> > > >
> > > > Last but not least, our methods can also work with random variable rewards discussed in A2 with a small adaption because of the difference in MDP.
> > > >
> > > >
> > > > >Q2: I'm also not so sure about the assertion that MDPs can't have stochastic rewards.
> > > >
> > > > A2: We get what the reviewer means. They present a case where the reward is modeled as a random variable.
> > > >
> > > > There are two choices to build the environments: one is to include the information of win or not in the next state $s^\prime$, and the other one is not to do that. If the first one is the case, our methods can still work as we input $s^\prime$ to the distributional reward  critic as we proposed. If the second one is the case, we do not think that is a fair game for our methods. First, the random variable reward is not the case we consider according to the definition of MDP in SR and RE papers. The rewards of all the environment settings considered in SR and RE are not random variables. Second, it is unfair for our methods not to know the rewards are random variables. It is the case that computing the expectation of rewards by RE just fits the betting game without perturbations. As our methods are to deal with perturbations, the expectation might be turned negative, breaking the optimal policy, which cannot solved by RE as well. Third, our methods can also work with knowing rewards are random variables (under another MDP setting). Assuming the environment is clean, what we should do is to compute the expectation of the reward label distribution predicted by our distributional reward critic, which works the same as RE. Besides, when the environment is perturbed and we have an assumption of the perturbed distribution or the clean one, we can decide how to interpret the output from our distributional reward critic with reference to the pre-knowledge, which cannot be realized by RE because it is just to fit the expectation.
> > > >
> > > >
> > > > >Q3: My point really was that the proof amounts to "under unrealistically perfect conditions, the approach works", which isn't surprising IMO and just over-mathsifies the paper
> > > >
> > > > A3: That makes sense. We are open to either simplifying/removing the proof in the Appendix or stating the theorem clearly and informally in the paper.
> > > >
> > > >
> > > > >Q4: This isn't true for episodic tasks. Consider an environment where an agent can choose to continue or terminate the task. If all rewards are positive, the optimal policy is to continue, but if they're shifted to negative values by setting $\omega_1<0$
> > > >  then it's optimal to terminate the task.
> > > >
> > > > A4: We get what the reviewer means. For non-episodic tasks, we usually regard the episode termination as entering absorbing states with only zero rewards to make the notations of consistent for episodic and non-episodic tasks. If that is the case, the value of $\omega_1$ does not influence the optimal policy. If not, we agree with what the reviewer suggested. The sufficient condition for RE to work is $\mathbb{E}(\tilde{r})=\omega_0\cdot r +\omega_1(\omega_0 > 0, (\omega_0\cdot r +\omega_1)\cdot r >0)$ to make sure the signs of the expected perturbed rewards are not flipped. We revise the paper regarding this.
> > > >
> > > > Again, we thank the reviewer for their insightful ideas and hope their concerns are addressed properly. If they are satisfied with the rebuttal, we kindly request the reviewer to raise their score. If not, we would be happy to continue the discussion during the rebuttal.

---

> > > > > ### Comment · Reviewer_XsR2 · 2023-11-22
> > > > >
> > > > > Thanks for your further comments, I think we're on the same page now. At this point I think I just need to chat with the other reviewers about the case where the reward is modelled as a random variable. I feel like the LLM scenario in the intro is not a particularly good motivating example, since I would argue that the reward *is* a random variable in that case. (The reward isn't corrupted per se; it's just that different people have different tastes.) If, say, four people are mildly pleased with the output of an LLM, but one person is deeply offended, I'm not sure it's a good idea to go with the majority reward; that seems unsafe. If, on the other hand, one user accidentally hits the wrong button when scoring, then the approach seems more applicable. At the very least, I agree with Reviewer wJ8T that "the paper should be upfront about the potential bias introduced by ignoring the stochasticity of the rewards".

---

> > > > > > ### Author Response · Authors · 2023-11-22
> > > > > >
> > > > > > We thank the valuable reply from the reviewer. Regarding MDP with stochastic clean rewards and LLM as a motivating example, we want to add some potentially useful information for the reviewer’s further reference.
> > > > > >
> > > > > > For MDP with stochastic clean rewards (bandit MDP), in our honest opinion, the question should be asked in our definition of MDP (control MDP) instead of the applicable scenarios of the method. We can not expect a method to work in the MDP that does not accord with their definition. Besides, our definition of MDP follows SR and RE methods and it is also what people usually define.
> > > > > >
> > > > > > In the definition of bandit MDP, people would always define the reward as a random variable instead of a transformation to a scalar as control MDP. There is no need to request a method to work in two kinds of MDP at the same time as people always have knowledge of the characteristics (e.g. what form of the reward function it is) when they are in an MDP. Besides, our methods can be adapted to bandit MDP just after a small adaptation.
> > > > > >
> > > > > > For LLM, the example given by the reviewer is a good one. We have different methods for different goals. As our methods are to deal with perturbations, we can not expect them to realize other goals. If we want to consider safeness, we can substitute softmax with temperature softmax and even reweight the distribution according to our goals and assumptions, meaning we can do almost everything as we get the label distribution. That corresponds to why we propose LLM as a motivating example exactly. Our methods can predict the distribution of discrete labels explicitly, which can not be realized by the baseline methods, meaning our methods can do more than them.
> > > > > >
> > > > > > We want to convey we can always think about a case to break a method by breaking its assumption, which is the same for all the methods. However, that case might even not be its target. The examples of the betting game and the LLM case break our methods by breaking our assumption (mode-preserving), and they are of different MDPs and different goals. As we propose in the paper, our goal is to deal with perturbed rewards brought by perturbations in control MDPs and our methods work under the mode-preserving assumption.
> > > > > >
> > > > > > To summarize, we hope the reviewer to focus on our settings (control MDPs) and our goals (handling perturbations) instead of the untargeted cases. Besides, our methods can be easily adapted to bandit MDPs and for other goals as requested. At last, we promise to add a separate part to discuss the adaptation of our methods in the paper.

---

> ### Comment · Reviewer_XsR2 · 2023-11-17
> **Stochastic rewards**
>
> Thanks very much for your reply!
>
> > Otherwise, it even does not accord with the definition of MDP as the same state-action pairs can generate different rewards.
>
> Correct me if I'm wrong, but isn't the perturbed reward $\tilde{R}(s, a)$ stochastic? I thought that the main assumption in the problem setting was that the same state-action pairs *could* generate different rewards. Is the assumption that the clean rewards are deterministic but the perturbed rewards aren't? If so, what's the justification for this?
>
> Edit: I'm also not so sure about the assertion that MDPs can't have stochastic rewards. Even in Sutton and Barto's classic *Reinforcement Learning: An Introduction*, they state "In general, reward signals may be stochastic functions of the state of the environment and the actions taken" (Chapter 1).

---

### Official Review · Reviewer_m2dC · 2023-11-03

**Soundness:** 3 good
**Presentation:** 3 good
**Contribution:** 2 fair
**Rating:** 6
**Confidence:** 3

**Summary:**

This paper addresses the problem of performing reinforcement learning (RL) when the true rewards have been perturbed. Like previous works, the paper considers problems where the reward function is perturbed in a specific way: the range of the reward function is discretized into equal-width subintervals, which are then perturbed according to a row-stochastic "confusion matrix". In particular, this work assumes that the confusion matrix is mode-preserving in that "true reward is the most frequently observed class after perturbation" (see abstract). Under this condition, a method is proposed that learns a classifier mapping state-action pairs to bins corresponding to observed modes and uses it to recover the true reward. These recovered rewards are then used when training the user's RL algorithm of choice. Versions of the proposed method are provided for (a) when the number of subintervals and reward range is known, (b) when the reward range but not the number of subintervals is known, and (c) when neither is known. The proposed method improves on existing works that either apply only to reward perturbations that preserve the order of the true rewards or require that the true rewards take only finitely many values. Experimental results are provided indicating superior performance over existing methods on a variety of tasks.

**Strengths:**

The paper addresses an interesting and important problem that is likely of interest to the community. The proposed method generalizes previous work in the area (see summary above) and provides a practical scheme for performing RL with a certain class of perturbed rewards. Though the class of perturbations assumed is potentially restrictive (see weaknesses section below), the method proposed for training a classifier and recovering the true reward is clever and appears to be novel. The experimental evaluation is quite extensive and indicates superior performance to existing methods on the perturbed-reward environments tested.

**Weaknesses:**

Two important weaknesses of this paper include the following. If these issues can be clarified, it will be easier to more accurately judge the significance of this work.
1. Though the proposed method avoids some of the drawbacks of previous works, such as directly estimating the confusion matrix or the true reward function, or assuming order-preservation of the perturbation, the mode-preserving condition seems quite restrictive. Mode-preserving perturbations are not a generalization of order-preserving perturbations, since the latter may not be order-preserving. Also, methods that learn the confusion matrix may be more widely applicable (though more computationally expensive), since the perturbations can be quite general. Without further motivation of the mode-preserving condition and its advantages over other conditions imposed in the literature, the broader impact of this work is difficult to judge.
2. The experimental comparisons are extensive, but there are some issues regarding the fairness of comparisons with previous methods as well as their interpretability. Due to the lack of order-preservation mentioned at the end of the first paragraph of Sec. 5.3, the results in Fig. 5 and 6 may be unfair to the RE method. Comparisons on problems that enjoy conditions to which all methods are applicable would provide a fairer comparison. In addition, though the proposed DRC and GDRC methods show better performance than previous methods on a number of problem instances, they decidedly underperform on many others. Further discussion of this discrepancy is warranted to clarify when the proposed methods should be preferred. Finally, the impact of $n_o$ studied in Sec 5.2 is an important issue, but the meaning of Fig. 3 is difficult to interpret. This leaves the question of how to choose $n_r$ open, which is a key component to the applicability of GDRC, in particular.

**Questions:**

* the proposed method replaces the computational effort of estimating the confusion matrix with the computational cost of training a classifier estimating the bin to which the reward at a given state-action pair belongs; can you comment on how these two costs compare?
* does the classifier trained in line 7 of Alg. 1 -- which is called the "distributional reward critic" -- correspond to any specific action value function, like the standard critics used in actor-critic algorithms? if not, it is a little misleading to call it a critic.
* what is the main difficulty in proving Theorem 1? it seems like it should follow from a straightforward application of the law of large numbers.
* what should the reader take away from Fig. 3? it is currently tough to interpret.
* at the end of the **Algorithms** subsection of Sec. 5.2, it says the rewards are averaged over "10 seeds and 20 random trials"; independent trials are usually each associated with one seed -- can you clarify what you mean?
* in Fig. 6, why do you think DRC's performance improves as noise improves?

---

> ### Author Response · Authors · 2023-11-17
> **(1/3) Rebuttal by Authors**
>
> We thank the reviewer for their valuable review. Here are our answers to the questions:
>
> >Q1: Order-preserving perturbations might not be mode-preserving and what the motivation of the mode-preserving assumption is.
>
> A1: Order-preserving is necessary for RE to work but is not sufficient. After analyzing all the methods in detail, we find the assumption of RE is $\mathbb{E}(\tilde{r})=\omega_0\cdot r +\omega_1(\omega_0 > 0)$, a positive affine transformation. That means RE might not work if order-preserving holds although mode-preserving might not hold as well. Here we clarify the difference between mode-preserving and order-preserving. Mode-preserving and order-preserving are two kinds of concepts although they look similar. Mode-preserving is the assumption of our methods and order-preserving is a property assumptions might necessitate. The assumption of RE is positive affine transformation instead of order-preserving. It is not that clear which method is more broadly applicable with different assumptions. Then we find positive affine transformation necessitates order-preserving, making RE not work under non-order-preserving cases as we discuss in the last paragraph of Page 2. However, mode-preserving does not necessitate order-preserving, which means our methods can work in cases where the reward order is totally broken, and then the optimal policy gets changed as long as mode-preserving holds, demonstrating the larger applicability and capability of our methods. That is the reason we have a separate column for order-preserving in Table 1 to help understand the applicable scenarios of different methods. To ensure there is no confusion, we change the term from "order-preserving" to "optimal-policy-unchanged", which is comparable to the positive affine transformation and is easy to tell the applicable scenarios as well. Besides, mode-preserving is not created by us, which is the assumption of SR as shown in Table 1. When recovering the confusion matrix, they use majority voting to decide the true reward of the samples for a state-action chunk while assuming mode-preserving.
>
> One more thing to notice is that the forms of $\tilde{R}$ of the continuous perturbation functions (e.g. Gaussian perturbations) people pay attention to in RE necessitate mode-preserving although the space is continuous, which shows mode-preserving is a weak assumption in the cases people are interested in. With mode-preserving, our methods win over the baseline methods under both GCM perturbations and continuous perturbations, which is the strongest evidence that mode-preserving is a weak assumption and our methods are broadly applicable.
>
> We clarify the motivation of mode-preserving in the last paragraph of Section 3 of the revised paper in blue.
>
> >Q2: Methods that learn the confusion matrix may be more widely applicable since the perturbations can be quite general.
>
> A2: Although the general confusion matrix (GCM) perturbation is general, the method (SR) learning the confusion matrix is not widely applicable.
>
> As we analyze in Table 1 and the last paragraph of Page 4, SR requires some unrealistic information about GCM settings, making it inapplicable to perturbations without the concept of a confusion matrix like continuous ones. Besides, although a correctly recovered confusion matrix tells us how perturbations are applied, it is not the best solution to have the fixed substitute reward for a corresponding observed reward using $\hat{R}=C^{-1}\cdot R$ because they might be perturbed from different clean rewards, which ignores the valuable state-action information and could even cause significant variance compared with the one caused by perturbations. Last but not least, SR discretizes the state-action space and regards the clean rewards of a given state-action chunk as the same without any assumption of the smoothness / Lipschitz-ness of rewards, making itself incapable of dealing with the environments with large state-action space and complicated reward functions.
>
> >Q3: The results in Figures 5 and 6 may be unfair to the RE method. Comparisons on problems that enjoy conditions to which all methods are applicable would provide a fairer comparison.
>
> A3: As mentioned in A1, a positive affine transformation is the assumption of RE instead of order-preserving and mode-preserving is what we need to work under all the perturbations people are interested in of both RE and SR. For Figures 5 and 6, the generation of GCM perturbations does not ensure any condition including mode-preserving as we say in the Perturbations paragraph of Page 7, meaning they are fair comparisons for all the methods. Additionally, we have evaluated the methods under continuous perturbations of RE with a positive affine transformation and found that our method still performs better although they are even not our targets, as discussed in Section 5.4.

---

> > ### Author Response · Authors · 2023-11-17
> > **(2/3) Rebuttal by Authors**
> >
> > >Q4: While the proposed DRC and GDRC methods show improved performance on several problem instances, they underperform on many others.
> >
> > A4: Our methods generally win over the baseline methods in all kinds of perturbation settings (e.g., environments, noise ratios, and reward discretizations) except for HalfCheetah with low noise ratios under GCM perturbations. Even under non-GCM perturbations (continuous ones), GDRC still has an edge over RE.
> > For HalfCheetah, we analyze the underperformance of our methods and provide some feasible solutions. In the last paragraph of Section 5.3, we have detailed discussions by analyzing the metrics of DRC in Hopper, DRC in HalfCheetah, and GDRC in HalfCheetah shown in Figure 10. Correspondingly, we have proposed potential solutions in the first paragraph of our Future Work section. As this is the first work introducing the distributional reward critic methods (DRC and GDRC) to address perturbed rewards and because of space constraints, we leave further improvement in the future work.
> >
> > >Q5: The impact of $n_o$ studied in Section 5.2 is crucial, but the meaning of Figure 3 is difficult to interpret.
> >
> > A5: Yes, choosing $n_o$ is an important problem to solve, which is the transition step from DRC to GDRC. There are two forces in Figure 3: one is the reconstruction error turns zero when $n_o$ is a multiple of $n_r$; the other one is overfitting becomes worse as $n_o$ increases because samples are not infinite.
> > For small $n_r$ like 5 or 10, the performance of DRC reaches the best when $n_o=n_o$, but it is almost always worse when $n_o$ is a multiple of $n_r$. For large $n_r$ like 20, the performance of DRC decreases even when $n_o$ is smaller than $n_r$. Both of the two phenomena above from Figure 3 support what we proposed by two forces before.
> >
> > In the method section before any experiments, we discuss the tradeoff between the two forces in the first paragraph of Page 6 and decide to choose when $n_o=n_r$, which is further proved correct in Section 5.2. Moreover, we discuss some potential solutions to further weaken the assumption of unlimited samples in the Future Work section.
> >
> > We clarify what we want to convey in the second paragraph of Section 5.2 of the revised paper in blue.
> >
> > >Q6: Comment on how the costs of estimating the confusion matrix and the reward distribution compare.
> >
> > A6: For time complexity, the cost difference between the two methods is negligible compared to the inherent training expenses of the critic and actor. For sample complexity, the samples we need to make the two methods work increase as $n_r$ increases. In the case of the known reward range and $n_r$, the sample complexity of SR and DRC is the same as the two methods both need enough samples to extract the mode reward. In the case of unknown discretization of the perturbation, the sample complexity of SR and GDRC is incomparable as SR cannot apply in the unknown cases.
> >
> > >Q7: Does the classifier trained in line 7 of Algorithm 1, referred to as the "distributional reward critic," correspond to any specific value function?
> >
> > A7: No, it is a reward critic to predict rewards instead of values, but the inputs are also state-action pairs. As we turn the regression problem into a classification one, it is a distributional reward critic. If you go to Figure 8, the pipeline of our methods from a high perspective, you can tell the classifiers are processed in P2, which is a different process from Value Fitting.
> >
> > >Q8: What is the main difficulty in proving Theorem 1? It seems like it should follow from a straightforward application of the law of large numbers.
> >
> > A8: The primary challenge lies in the fact that an input (e.g., an image) has a definite label in typical classification problems. However, in our perturbed environment, samples of the same state-action pair can have varying labels. It is difficult to prove the minimum of the loss function for all the samples directly so we wisely pick the samples of a state-action pair and prove the minimum of the loss function for these samples. Our goal is to demonstrate that the output of the trained distributional reward critic corresponds to the distribution of perturbed reward labels by using Cross-Entropy (CE) loss. Indeed, the law of large numbers applies to the assumption we say as infinite samples.
> >
> > >Q9: What should the reader take away from Figure 3?
> >
> > A9: As mentioned in A5, the tradeoff between the two forces, the relation between the reconstruction error and $n_o$ and overfitting issues as we increase $n_o$, we propose in the first paragraph of Page 6 under the method section is proved in Section 5.3, meaning our strategy to choose $n_o$ concluded from the simulation environments is correct in experimental environments.
> >
> > >Q10: What does the use of 10 seeds and 20 random trials imply?
> >
> > A10: We train the agents using 10 different seeds. For each seed, we run the agent for 20 episodes after the training for evaluation.

---

> > > ### Author Response · Authors · 2023-11-17
> > > **(3/3) Rebuttal by Authors**
> > >
> > > >Q11: In Figure 6, why do you think contributes to the improvement in DRC's performance as noise levels increase?
> > >
> > > A11: This is because any method handling perturbations inevitably suffers from a performance decrease under perturbations compared to the trained agent using clean rewards. In less perturbed environments, the methods might even underperform compared to using perturbed rewards directly. However, the methods gradually outperform the direct use of perturbed rewards and the improvement gets more and more significant as the noise ratio increases. It is the same case for our methods (DRC).

---

> > > > ### Author Response · Authors · 2023-11-20
> > > >
> > > > Again, we thank the reviewer for their valuable review and hope their concerns are addressed properly. If they are satisfied with the rebuttal, we kindly request the reviewer to raise their score. If not, we would be happy to continue the discussion during the rebuttal.

---

> > > > > ### Comment · Reviewer_m2dC · 2023-11-23
> > > > >
> > > > > Thanks to the authors for their response, which helped to clarify some of my questions. However, I remain concerned about the motivation and justification for the restrictive mode-preserving condition and still lack a clear picture of the types of problems for which this is a reasonable assumption. In addition, I am not convinced that the proposed method truly generalizes existing approaches, as claimed in the abstract. Indeed, the response leads me to believe that, just as the proposed method may work for problems where order preservation fails, alternative methods may work when mode preservation fails. I will therefore keep my score.

---

> ### Author Response · Authors · 2023-11-23
>
> We thank the valuable reply from the reviewer. We just want to have a quick clarification for the reviewer’s further reference.
>
> >Point 1: I remain concerned about the motivation and justification for the restrictive mode-preserving condition and still lack a clear picture of the types of problems for which this is a reasonable assumption.
>
> Reply 1: We are happy to tell why we choose mode-preserving as the assumption of our methods. We want to design a method that can deal with a general perturbation. Then we generalize the perturbation in SR to form our generalized confusion matrix (GCM) perturbations, which can cover any reward perturbation as we propose in Section 3.2. In the step of deciding on the assumption of our methods, there are three factors contributing to our decision. First, the assumption of RE, optimal-policy-unchanged, is very strict and the optimal policy after this perturbation can change easily. Second, SR seems to work under this perturbation as long as its assumption works, which is mode-preserving. Third, what GCM perturbations do is transform the discrete bias caused by perturbed labels into a continuous space. In the field of handling perturbed (noisy) labels, mode-preserving is the most common assumption. Therefore, we choose mode-preserving as our assumption and we do not find a better choice than it under GCM perturbations.
>
> Mode-preserving is not a restrictive assumption. Mode-preserving is a comparable assumption of optimal-policy-unchanged at least, where the former cares about mode and the latter cares about expectation. Besides, assuming the mode of perturbed labels under GCM perturbations makes more sense as we say in the second point above. Even for the continuous perturbations people pay attention to in RE, mode-preserving weakly holds, explaining why our methods work there. What’s more, tuning LLMs is also an appropriate scenario to apply our methods for handling noisy labels. As we discuss with other reviewers, our methods can be even adapted to customize to different needs in LLMs, meaning the assumption can be anything besides mode-preserving.
>
>
> >Point 2: I am not convinced that the proposed method truly generalizes existing approaches, as claimed in the abstract.
>
> Reply 2: We do not say our methods generalize existing approaches in the abstract. What we convey is that our GCM perturbations generalize the existing perturbations and the baseline methods do not work because of either unreasonable needed information or poor performance.
>
>
> >Point 3: The proposed method may work for problems where order preservation fails, alternative methods may work when mode preservation fails.
>
> Reply 3: Yes, that is what we conveyed in the previous rebuttal. It is impossible to decide the better one between assuming the expectation and assuming the mode directly. However, we do not choose to assume the expectation because it ensures the optimal policy does not change regarding the expectation of the perturbed rewards, which greatly restricts the influence that can be caused by perturbations. To tell it simply, such perturbations can only cause variance but no bias to influence the optimal policy. Another point to mention is as we always say in the previous rebuttal and Reply1 above, mode-preserving weakly holds under the continuous perturbations people are interested in, which is not an unreasonable point. Generally speaking, when we consider a perturbation only causing variance, what people usually choose is something like Gaussian perturbation with one mode instead of the perturbation of a multimodal distribution, meaning those perturbations are more realistic to some extent.

---

### Meta-Review · Area_Chair_UTJN · 2023-12-06

**Metareview:**

The paper investigates the problem of RL under reward perturbation, considering a case of unknown arbitrary perturbations that discretize and shuffle reward space. Under the assumption about the mode-preserving property of reward perturbations, the paper proposes a new method to learn the reward distribution and train an RL agent from the mode of the distribution. The reviewers agreed that the paper considers an important reinforcement learning setting under reward perturbations, and some reviewers appreciated new ideas in the proposed method. However, the reviewers pointed out several weaknesses and shared a common concern that the paper makes a strong and restrictive assumption about the mode-preserving property of reward perturbations. We want to thank the authors for their detailed responses; the paper was discussed among all the reviewers, considering the responses. Based on the raised concerns and follow-up discussions, unfortunately, the final decision is a rejection. Nevertheless, this is exciting and potentially impactful work, and we encourage the authors to incorporate the reviewers' feedback when preparing a future revision of the paper.

**Justification For Why Not Higher Score:**

The reviewers shared a common concern that the paper makes a strong assumption about the mode-preserving property of reward perturbations, which is restrictive and not very practical. The paper was discussed among all the reviewers, and they are inclined toward a rejection decision.

**Justification For Why Not Lower Score:**

N/A

---

### Decision · Program_Chairs · 2024-01-16

Reject